JCB Journal of Cell Biology

# Recycling of cell surface membrane proteins from yeast endosomes is regulated by ubiquitinated Ist1

Kamilla M.E. Laidlaw[1], Grant Calder[2], and Chris MacDonald[1]

**Upon internalization, many surface membrane proteins are recycled back to the plasma membrane. Although these endosomal trafficking pathways control surface protein activity, the precise regulatory features and division of labor between interconnected pathways are poorly defined. In yeast, we show recycling back to the surface occurs through distinct pathways. In addition to retrograde recycling pathways via the late Golgi, used by synaptobrevins and driven by cargo ubiquitination, we find nutrient transporter recycling bypasses the Golgi in a pathway driven by cargo deubiquitination. Nutrient transporters rapidly internalize to, and recycle from, endosomes marked by the ESCRT-III associated factor Ist1. This compartment serves as both "early" and "recycling" endosome. We show Ist1 is ubiquitinated and that this is required for proper endosomal recruitment and cargo recycling to the surface. Additionally, the essential ATPase Cdc48 and its adaptor Npl4 are required for recycling, potentially through regulation of ubiquitinated Ist1. This collectively suggests mechanistic features of recycling from endosomes to the plasma membrane are conserved.**

## Introduction

The plasma membrane (PM) of eukaryotic cells hosts a variety of functionally diverse proteins that play critical cellular roles, often detecting extracellular signals and ensuring appropriate responses to the environment. As such, surface membrane proteins represent by far the largest class of molecular therapeutic targets (Santos et al., 2017). Surface proteins are regulated by intracellular transport pathways, relying on different vesicle trafficking steps transiting the secretory pathway to the PM (Stalder and Gershlick, 2020) or endocytic regulation following their initial localization at the surface (Sigismund et al., 2021). Many of these mechanisms are evolutionarily conserved, with fundamental modes of regulation uncovered and explained in the budding yeast *Saccharomyces cerevisiae*.

Surface proteins can be downregulated through endocytosis and trafficking to the lysosome (vacuole in yeast), including the specific downregulation of receptors and transporters recognizing ligands and substrates, respectively (Davis et al., 1993; Séron et al., 1999). More general cargo endocytosis can occur en masse, such as yeast cells triggering large-scale downregulation of surface cargoes in response to nutrient stress (Müller et al., 2015; Laidlaw et al., 2021). Endocytosed surface cargoes that retain a ubiquitination signal are targeted through the multivesicular body (MVB) pathway, packaged into intraluminal vesicles (ILVs), and then delivered to the vacuole for degradation (Urbanowski and Piper, 2001; Katzmann et al., 2001).

Ubiquitinated cargoes are recognized by the endosomal sorting complex required for transport (ESCRT) complexes, with ESCRT-0 and ESCRT-I abundant in ubiquitin-binding domains (Shields et al., 2009). ESCRT-III subunits are recruited to the endosomal/MVB membrane, where they polymerize and drive formation of vesicles budding into the endosome lumen (Hanson et al., 2008; Saksena et al., 2009; Wollert et al., 2009). Filaments created through polymerization of Snf7, modulated by other ESCRTs and the Vps4 ATPase, drive membrane deformation (Shestakova et al., 2010; Adell et al., 2017; Maity et al., 2019; Tang et al., 2015; Pfitzner et al., 2020). Nutrient transporters in yeast are useful reporters for endosomal trafficking, such as the amino acid transporters Mup1 and Fur4, as their ubiquitination and ESCRT-dependent trafficking to the vacuole can be triggered by addition of substrate (Hein et al., 1995; Keener and Babst, 2013; Isnard et al., 1996). The E3-ligase Rsp5 and its substrate specific adaptors are largely responsible for cargo ubiquitination and vacuolar-sorting events (MacDonald et al., 2020; Sardana and Emr, 2021).

Surface activity of internalized PM proteins is also regulated by recycling routes back to the PM (MacDonald and Piper, 2016). Recycling in yeast appears less complex than in mammalian cells, but the division of labor between certain pathways, and even organization of the endosomal system, is not fully understood (Day et al., 2018; Laidlaw and MacDonald, 2018; Ma and

[1]York Biomedical Research Institute and Department of Biology, University of York, York, UK;   [2]Imaging and Cytometry Laboratory, Bioscience Technology Facility, Department of Biology, University of York, York, UK.

Correspondence to Chris MacDonald: chris.macdonald@york.ac.uk.



Burd, 2019). Retrograde recycling of yeast synaptobrevin orthologs, Snc1 and Snc2 (Protopopov et al., 1993), via the TGN has been extensively studied. Localization of Snc1 is polarized with concentration to the bud-tips of emerging daughter cells and the shmoo protrusions induced upon response to mating factor; this polarization relies on postendocytic recycling (Valdez-Taubas and Pelham, 2003). Snc1 recycles via the TGN through multiple pathways involving different machinery, such as retromer, Snx4-Atg20 (Lewis et al., 2000; Hettema et al., 2003; Ma et al., 2017), and other factors, including phospholipid flippases, Rcy1, and Ypt31/32 (Galan et al., 2001; Hua et al., 2002; Chen et al., 2005; Furuta et al., 2007; Hanamatsu et al., 2014). Retrograde recycling of Snc1 also requires its ubiquitination to facilitate interaction with endosomally localized α subunit of COPI vesicle coatomer complex, COPI (Xu et al., 2017). This is an intriguing observation, as ubiquitinated Snc1 can be incorporated into the MVB pathway and sorted to the vacuole during stress conditions (Chen et al., 2011; MacDonald et al., 2015). However, this suggests that under normal conditions, the primary role of Snc1 ubiquitination is for recycling and implies different ubiquitin receptors dedicated to distinct pathways operate, even within the yeast endosomal network.

Unlike Snc1, deubiquitination of other cargoes, like nutrient transporters that are typically sorted to the vacuole upon ubiquitination, appears to have the opposite effect. Directing catalytic activity of deubiquitinating enzymes (DUb) to cargo, either directly or indirectly via Rsp5 or ESCRT proteins (Stringer and Piper, 2011; MacDonald et al., 2012a; MacDonald et al., 2012b), antagonizes cargo degradation and triggers recycling by default. To characterize this pathway, a genetic screen using a DUb-fused version of the receptor Ste3 (Ste3-GFP-DUb) was performed to reveal 89 factors that are required for this recycling (MacDonald and Piper, 2017), most of which were validated with additional assays for Tat2 (Johnson et al., 2010) and FM4-64 (Wiederkehr et al., 2000) recycling. Null mutants of retrograde machinery, like retromers Snx4/41/42 and Ere1/2 (Seaman et al., 1998; Hettema et al., 2003; Shi et al., 2011), were not identified by the screen and recycle FM4-64 at similar rates to WT cells. Furthermore, cells harboring a mutant allele of SEC7 with abrogated trafficking through the Golgi also exhibit efficient DUb-triggered recycling. These data suggest the recycling of DUb-fused cargoes is predominantly distinct from retrograde recycling of Snc1, but there is likely overlap between pathways as some endosomal machinery that Snc1 relies on was shown to be required for Ste3-GFP-DUb recycling (e.g., Rcy1, Cdc50-Drs2, and Ypt31/32).

This genetic screen also implicated Ist1 in the yeast endosomal recycling pathway triggered by cargo deubiquitination. Ist1 shares structural homology with other ESCRT-III subunits and contributes to the efficiency of MVB sorting (Dimaano et al., 2008; Rue et al., 2008; Xiao et al., 2009; Frankel et al., 2017; Buono et al., 2016; Pfitzner et al., 2020). Ist1 interacts with other ESCRT-III subunits and Vps4, the AAA-ATPase required for MVB sorting and disassembly of ESCRT polymers (Babst et al., 1997; Babst et al., 1998; Babst et al., 2002; Nickerson et al., 2006; Azmi et al., 2006). Ist1 regulation of Vps4 is complex; even in vitro Ist1 can stimulate and inhibit Vps4 activity, and control depends on other ESCRT-III subunits (Tan et al., 2015).

Therefore, despite great strides in our understanding of ILV formation by ESCRT-III filaments and Vps4 (Han et al., 2017; Adell et al., 2017; Maity et al., 2019; Pfitzner et al., 2020), our understanding of the role(s) of Ist1 in the endosomal assemblage of ESCRTs in vivo is incomplete. Furthermore, high levels of Ist1 inhibit MVB sorting, and diverse cargoes are sorted more efficiently to the vacuole in ist1Δ cells (Dimaano et al., 2008; Jones et al., 2012). In addition to the negative regulation of Vps4, it may be that Ist1 also promotes an opposite-acting recycling pathway to the PM. In support of this idea, in vivo and in vitro studies show that unlike ESCRT-III polymers, which drive luminal vesicle formation, Ist1 polymerization, in combination with CHMP1B, exerts the opposite effect on endosomal membranes to generate tubulation of cytosolic protrusions (McCullough et al., 2015; Nguyen et al., 2020). Physiologically, this can be best rationalized by Ist1 promoting the recycling pathway by creation/fission of recycling tubules that return material back to the PM. Cellular work has suggested this occurs in collaboration with the ATPase spastin (Allison et al., 2013, 2017), but a reconstitution of the process in vitro shows ESCRT assemblies in combination with a mechanical pulling force are sufficient to drive scission (Cada et al., 2022 Preprint).

In this study, we present evidence that, as expected, the surface cargoes Snc1/2 mainly follow a trafficking route to the TGN labeled with the Arf-exchange factor Sec7. However, this trafficking appears to be distinct from the pathway used by nutrient transporters for methionine (Mup1) and uracil (Fur4). Upon addition of substrate, nutrient transporters rapidly internalize to compartments marked by Vps4 and Ist1, followed by delivery to the vacuole. However, using microfluidics to rapidly remove the substrate-induced ubiquitin-degradation signal, Mup1 recycles directly back to the PM from this Vps4 endosome population. These trafficking events are not observed to transit Sec7-labeled TGN compartments. This recycling pathway driven by cargo deubiquitination relies on the ESCRT-III associated protein Ist1. We provide initial evidence that Ist1 is ubiquitinated, and this is required for its ability to recycle cargoes back to the PM. We also reveal Npl4-Cdc48 is required for proper endosomal regulation of Ist1-mediated cargo recycling.

## Results

### Differential cargo recycling routes in yeast

The v-SNARE Snc1 is a well-established retrograde cargo that internalizes and recycles via the TGN back to the PM through multiple pathways (Lewis et al., 2000; Best et al., 2020). Snc1 ubiquitination is required for recycling (Xu et al., 2017), and the fusion of the catalytic domain of a DUb is sufficient to block recycling of GFP-tagged Snc1 and the paralog Snc2 (Fig. 1 A). In contrast, substrate-induced ubiquitination of the Mup1 (methionine) or Fur4 (uracil) permeases does not promote recycling and instead drives endocytosis and vacuolar degradation (Menant et al., 2006; Lin et al., 2008; Hein et al., 1995; Keener and Babst, 2013). Mup1-GFP localizes to the surface but is almost entirely sorted to the vacuole within 1 h after methionine addition. This ubiquitin-dependent sorting is blocked by DUb-fusion to Mup1-GFP (Fig. 1 B). Similarly, although Fur4 steady

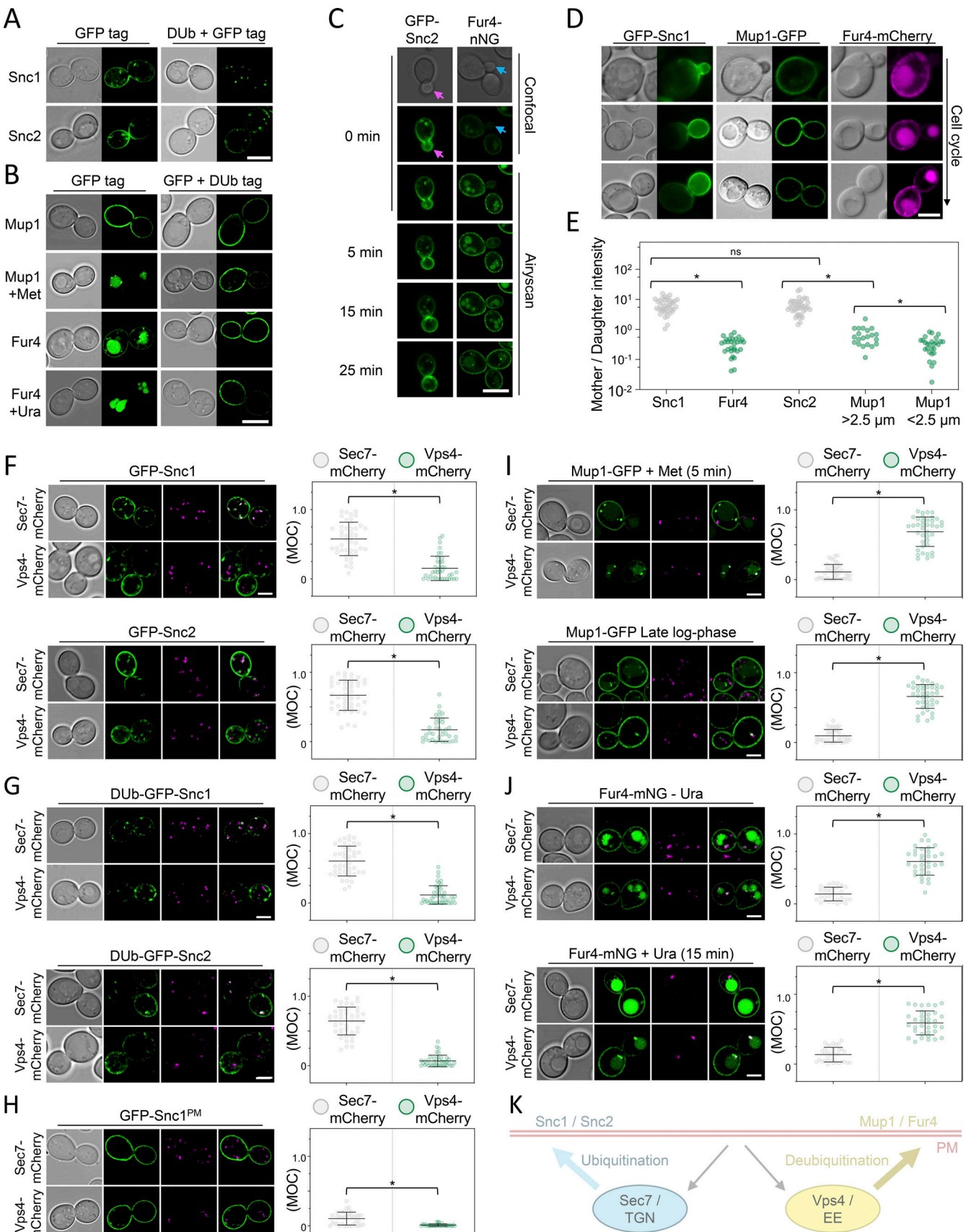

Figure 1. **Differential trafficking features of recycling cargoes. (A)** WT cells expressing Snc1 and Snc2 tagged with either GFP or a fusion of GFP with the catalytic domain of DUb UL36 (DUb + GFP) expressed from the *CUP1* promoter were imaged by Airyscan microscopy. **(B)** Mup1 and Fur4 expressed from their

endogenous promoters and fused to C-terminal GFP or GFP-DUb tags were imaged by Airyscan microscopy. Where indicated, 20 µg/ml methionine (+Met) and 40 µg/ml uracil (+Ura) were added to media 1 h prior to imaging. **(C)** Time-lapse microscopy of cells expressing GFP-Snc2 (left) or Fur4-mNG (right). **(D and E)** WT cells expressing fluorescently labeled Snc1, Mup1, and Fur4 were imaged, with example cell cycle stages depicted, and fluorescence in mother–daughter pairs quantified. *, P < 0.002. **(F–J)**. Indicated GFP tagged cargoes were imaged by Airyscan microscopy in Sec7-mCherry (upper micrographs) or Vps4-mCherry (lower micrographs) cells, with associated jitter plots of Mander's overlap coefficients (MOC). *, P < 0.0001 from unpaired *t* test. **(K)** Schematic summarizing distinct yeast endosomal recycling pathways. Scale bar, 5 µm.

state localization includes intravacuolar signal, any surface localized Fur4 is sorted to the vacuole in the presence of uracil, and this trafficking is blocked by DUb-fusion.

Retrograde recycling of Snc1/2 via the TGN is required for its polarized PM distribution (Lewis et al., 2000; Valdez-Taubas and Pelham, 2003; Hettema et al., 2003; Ma et al., 2017), with GFP-tagged Snc1/2 concentrated in budding daughter cells (Fig. 1, C and E). Unlike Snc1/2, fluorescently tagged Mup1 and Fur4 concentrate in the mother cell during budding (Fig. S1, A–E). This polarization is maintained throughout the cell cycle for cells expressing Fur4 but is only obvious in small cells (<2.5 µm diameter) expressing Mup1, with the levels increasing to roughly the same as mother cells later in the cell cycle. As expected, intracellular signal from retrograde cargoes Snc1/2 tagged with GFP colocalizes with the TGN marker Sec7-mCherry, with very little overlap with the endosomal marker Vps4-mCherry (Fig. 1 F). Furthermore, Dub-GFP-Snc1/2 fusions, which do not recycle efficiently, accumulate in Sec7-marked TGN compartments (Fig. 1 G). The majority of intracellular signal is contributed by recycling Snc1, and not first-pass molecules transiting the Golgi, as a mutant version of Snc1 (Snc1PM) with defective internalization (Lewis et al., 2000) exhibits very little intracellular signal, which maintains colocalization with Sec7-mCherry (Fig. 1 H). In contrast, intracellular Mup1-GFP, triggered by a 5-min exposure to methionine or by nutrient starvation experienced in cells grown to late log phase (MacDonald et al., 2012b, 2015), primarily internalizes to Vps4-mCherry, and not Sec7-positive TGN, compartments (Fig. 1 I). Similarly, Fur4-mNG that has significant intracellular signal irrespective of substrate presence colocalizes with Vps4 endosomes (Fig. 1 J). Although cargo-specific trafficking regulation might explain mother–daughter differences, taken with the opposing effects of enforced deubiquitination and differences in intracellular localization, we propose that endosomal recycling of nutrient transporters follows a distinct route than that used by Snc1/2 via the Golgi (Fig. 1 K).

### Nutrient transporters recycle from early endosomes

Although enforced deubiquitination of internalized nutrient transporters promotes their recycling to the surface, observing recycling of unmodified nutrient transporters is hampered by their proclivity for vacuolar sorting (Stringer and Piper, 2011; MacDonald et al., 2012a; MacDonald et al., 2012b; MacDonald et al., 2015). To overcome this, we optimized microfluidic exchange with continuous imaging to perform a substrate pulse, followed by washes and substrate-free chase to allow internalized cargo to recycle naturally (Fig. 2 A). Mup1-GFP was used for these experiments as it exhibits steady-state localization at the PM, and we found high (40 µg/ml) and low (2 µg/ml) levels of

methionine triggered accumulation of intracellular puncta, with high doses giving brighter, more obvious puncta (Fig. 2 B). The pulse-chase protocol resulted in accumulation of intracellular Mup1-GFP for 0–15 min followed by clearance of most signals after an additional 15 min, regardless of methionine pulse concentration. To avoid concerns about photobleaching or substrate-induced degradation, biochemical analysis was used to show that methionine pulse periods of 1 or 5 min, followed by substrate-free chase up to 60 min resulted in no increase in vacuolar delivery of internalized material (Fig. 2, C and D). All subsequent substrate-induced experiments were performed with <1 min methionine pulses. To confirm intracellular nutrient transporter signal emanated from the surface, we optimized photoconversion of surface localized Mup1-mEos and Fur4-mEos (Fig. 2, E and F). Coupling cargo photoconversion to microfluidic-induced recycling and time-lapse microscopy revealed a signal of intracellular Mup1 that has trafficked from the PM, which subsequently dissipates with similar kinetics to Mup1-GFP (Fig. 2 G and Video 1). We then followed the trafficking itinerary of internalized and recycled Mup1-GFP using 4D confocal Airyscan microscopy optimized for rapid acquisitions. Methionine-induced internalization tracked at 4–10-s intervals show substantial intracellular accumulations that colocalize with Vps4-mCherry, and not Sec7-mCherry, within the first few minutes (Fig. 3 A; and Videos 2 and 3). Additionally, Mup1-GFP recycling experiments were imaged over longer periods to show that Mup1 primarily traverses Vps4 endosomes, bypassing Sec7-mCherry compartments (Fig. 3, B and C; and Videos 4 and 5). We also tracked colocalization from time-lapse imaging experiments and found that Mup1-GFP almost exclusively colocalizes with Vps4-mCherry over time (Fig. 3, D–F).

### Ist1 is required for cargo recycling

Vps4, the ATPase involved in ESCRT-mediated ILV formation, marks a large and relatively static endosome reminiscent of Vps8 compartments (Day et al., 2018) in addition to a population of peripheral mobile compartments (Adell et al., 2017). The ESCRT-associated factor Ist1 colocalizes with Vps4 in both populations, in addition to potentially distinct endosomes (Fig. 4, A and B; and Video 6). As Ist1 was the only recycling factor from a blind genetic screen of ~5,200 mutants (MacDonald and Piper, 2017) that also interacts with Vps4 (Fig. 4 C), and its orthologs have been implicated in recycling (Allison et al., 2013; Frankel et al., 2017; McCullough et al., 2015), we hypothesized Ist1 might functionally define this population of recycling endosomes. We integrated the Ste3-GFP-DUb recycling reporter to confirm recycling defects of *ist1Δ* and *rcy1Δ* cells, in addition to quantifying their capacity for recycling Tat2 and FM4-64 (Fig. 4, D–F). Deletion of *ist1Δ* in cells expressing Mup1-GFP caused recycling

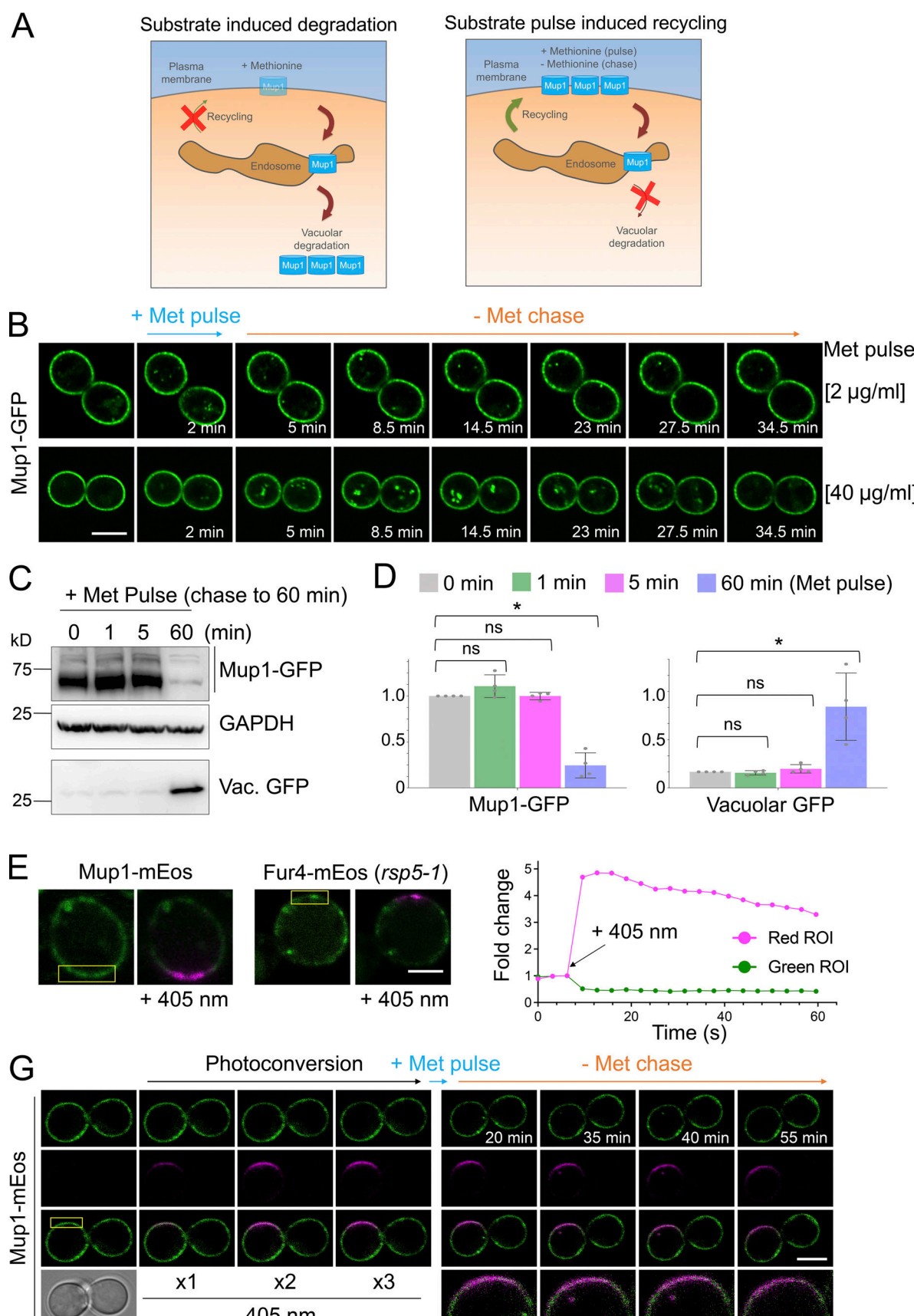

Figure 2. **Substrate-induced transporter recycling. (A)** Cartoon of substrate-induced degradation (left) and recycling (right) of Mup1 triggered by modulation of extracellular methionine. **(B)** Time-lapse Airyscan microscopy of cells expressing Mup1-GFP before and after 2-min methionine (2 µg/ml, upper and

40 µg/ml, lower) pulse-chase incubations. **(C)** Cells expressing Mup1-GFP were incubated with 20 µg/ml methionine for 0, 1, 5, and 60 min followed by three times washes and further incubation in SC-Met up to 60 min before lysates were generated and immunoblotted. **(D)** Quantification of average intensity of Mup1-GFP (left) and vacuolar processed GFP (right) from methionine pulse-chase experiments from C. *, P < 0.01. **(E and F)** Yellow regions from cells expressing Mup1-mEos and Fur4-mEos were exposed to 405-nm laser at 0.5% to photoconvert molecules (left) and mEOS fluorescence-tracked over time before (right). **(G)** Time-lapse microscopy of cells expressing Mup1-mEOS following three times pulse with 0.1% 405-nm laser followed by substrate-induced recycling stimulated by 2 µg/ml methionine for 30 s. Scale bar, 5 µm (white); 1 µm (yellow). Source data are available for this figure: SourceData F2.

defects, with a small population of Mup1-GFP retained in endosomes marked by Vps4-mCherry and distinct from Sec7-mCherry positive compartments (Fig. 4 G). Additionally, Mup1-GFP and Ist1-mCherry positive foci were captured trafficking towards to PM from fast imaging experiments (Video 7).

### Npl4-Cdc48 regulates Ist1 recycling

As the mammalian ortholog of yeast Ist1 has been implicated in polymerization and creation of cytosolic recycling tubules (McCullough et al., 2015; Allison et al., 2013; Allison et al., 2017), we reasoned this function could be conserved in yeast (Fig. 5 A). Vps4 is an obvious candidate AAA-ATPase for Ist1 depolymerization. However, $vps4\Delta$ cells exhibit only a marginal defect in FM4-64 recycling (MacDonald and Piper, 2017), and a mutant of Ist1 (Ist1$^{\Delta MIM}$) with significantly diminished Vps4-binding (Dimaano et al., 2008; Tan et al., 2015) rescues efficient FM4-64 recycling of $ist1\Delta$ cells (Fig. 5 B). Encouragingly, a similar mutation to mammalian Ist1 lacking Vps4-binding has recently been shown to function in membrane deformations that would be required for recycling (Cada et al., 2022 Preprint). Other Ist1 mutants (Ist1$^{K52D}$, Ist1$^{K74A}$, and Ist1$^{K135A}$) that are defective in regulating Vps4 (Tan et al., 2015) also recycle FM4-64 with no obvious defects when compared with WT Ist1 (Fig. S2 A). This suggests that Vps4 has no major or direct role in cargo recycling. This idea is supported by the observation that $vps4$ mutants, both $vps4\Delta$ nulls and cells expressing dominant negative Vps4$^{EQ}$ allele, that are defective in sorting the MVB cargo Cos5-GFP have no impact on recycling Ste3-GFP-DUb efficiently (Fig. 5 C). Collectively, this implies that the Vps4 ATPase required for ILV formation during cargo degradation can be functionally separated from Ist1-mediated cargo recycling from endosomes back to the PM.

Screening for recycling machinery did not identify any other ATPase candidates, but did implicate Npl4, an adaptor of the essential Cdc48 ATPase, as required for recycling (Fig. S2 B). We demonstrate that both Npl4 (Fig. 5, D and E) and Cdc48 (Fig. 5, F and G) are required to efficiently recycle Ste3-GFP-DUb and FM4-64 back to the PM. This led to our speculative model that the role of Cdc48-Npl4 in recycling was mediated via Ist1. In support of this idea, deletion of *IST1* in either an $npl4\Delta$ or a $cdc48$-2 mutant background did not result in further deficiency in recycling FM4-64 (Fig. 5, H and I), which would be predicted if Ist1 did function in the same pathway.

### Ist1 is ubiquitinated

We extended this hypothetical model that functionally connects Ist1 to Npl4-Cdc48 (Fig. 6 A), predicting that Ist1 ubiquitination would allow Cdc48 recruitment via the well-established Npl4-ubiquitin–binding motif (Wang et al., 2003). Although Ist1 is known to be turned over by the proteasome (Jones et al., 2012), it

has not been formally demonstrated to be ubiquitinated, likely owing to the abundance of Lys/Arg residues in its sequence, which represent 50/298 amino acids (Fig. 6 B). Commonly used enzymes upstream of mass spectrometry (MS), like Trypsin and LysC, cleave at Lys/Arg residues and would promote extensive digestion and reduce likelihood of identifying ubiquitin-modified peptides by MS (Swaney et al., 2010). In an effort to identify bona fide Ist1 ubiquitination sites, we created an endogenously tagged version of Ist1 with a C-terminal HA tag and a His$_6$ tag to allow purification under denatured conditions (Fig. 6 C). Purification from 2 l of cultures grown to log phase provides high levels of Ist1 enrichment by immunoblot, including protein species evident by Coomassie staining at this approximate molecular weight (Fig. 6 D). However, in-gel digestion of these bands followed by analysis using matrix-assisted laser desorption/ionization (MALDI) MS revealed these were not Ist1 (Fig. S3 A).

Instead, the purified eluates were prepared for liquid chromatography with tandem MS (LC-MS/MS) using an optimized protocol aimed at identifying ubiquitinated Ist1 peptides (Fig. 6 E). This included the sequential digest by ArgC followed by acidification and digestion with ProAlanase. Half the sample was also subjected to propionylation, which has previously been shown to increase the hydrophobicity and retention of lysine-containing peptides prior to MS (Drury et al., 2012). This revealed many previous yeast contaminants from denatured lysates (Fig. S3 B), thousands of unique peptides (Table S5), in addition to several distinct Ist1 peptides (Fig. 6 F). One spectrum matched as a ubiquitin remnant modified peptide (diGly), even at 5% false discovery rate (FDR). However, due to low intensity, the spectra could not be manually confirmed. Furthermore, the fact that Ist1 has 38 lysine residues spread across only 298 amino acids, almost double the average across the yeast proteome (Fig. S3 C), multiple redundant ubiquitination sites are possibly retained.

As an alternative approach, we performed denatured ubiquitome purification using an optimized strain expressing His$_6$ tagged ubiquitin (MacDonald et al., 2017) to reveal that ubiquitinated species of Ist1 can be observed (Fig. 6 G and Fig. S3 D). We believe this pull-down result represents ubiquitinated Ist1, as a repeat in $ist1\Delta$ mutant strain shows similar levels of purified material and nonspecific bands, but no Ist1 at its predicted molecular weight (Fig. 6 H). Collectively, this data suggests that Ist1 is ubiquitinated, and we provide initial evidence that minimally K135, which is highly conserved across evolution (Fig. S3 E), might serve as a ubiquitination site.

### Ubiquitinated Ist1 is required for recycling

Our model would predict that ubiquitination of Ist1 is required for recycling, promoting recruitment of Cdc48 via Npl4 (Fig. 6

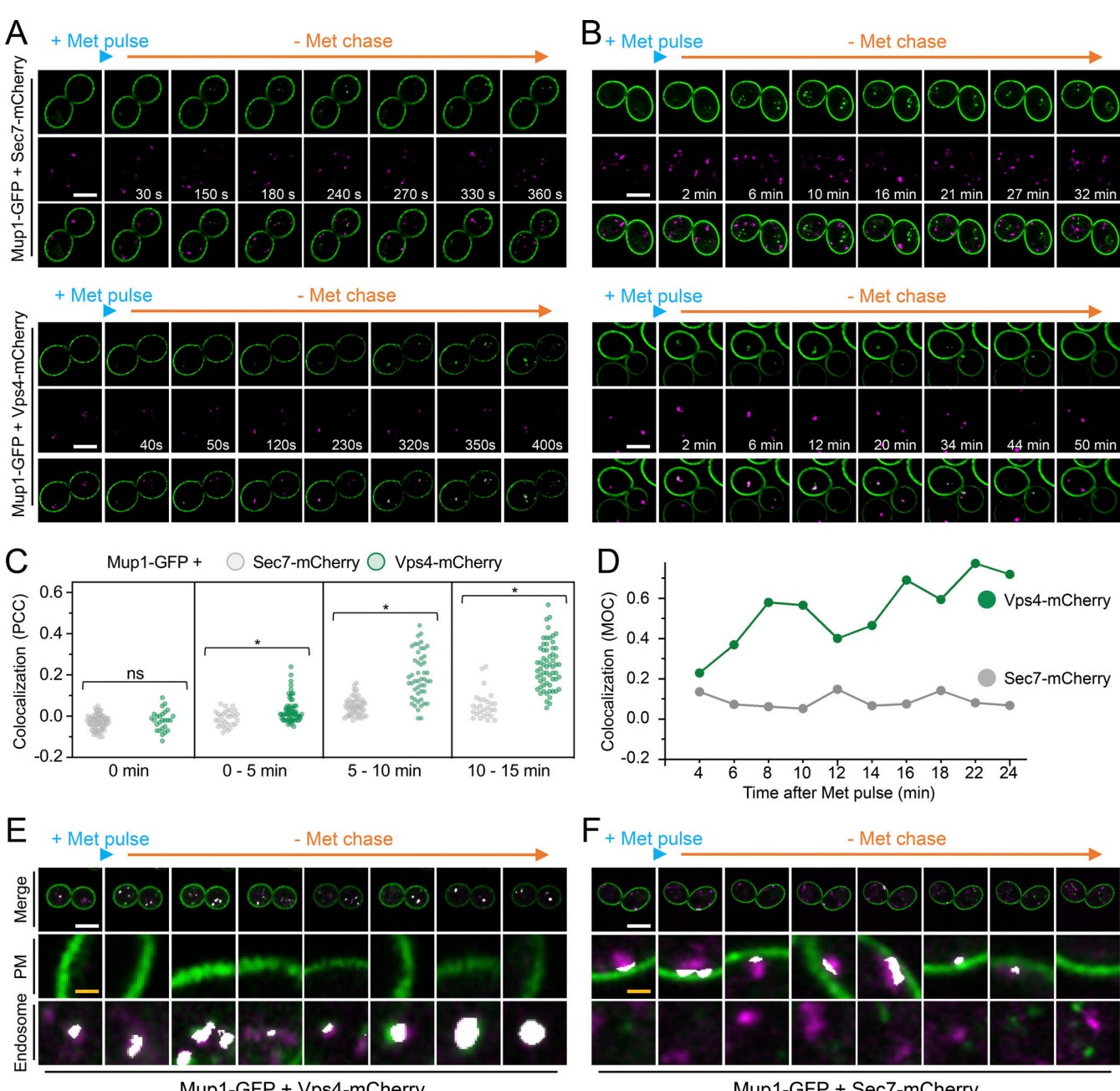

Figure 3. **Mup1-GFP recycling occurs from a Vps4-Ist1 endosome. (A and B)** 4D Airyscan microscopy of WT cells co-expressing Mup1-GFP and Sec7-mCherry (upper) and Vps4-mCherry (lower) following a 30-s 20 µg/ml methionine pulse and subsequent SC-Met chase period over short 2–4 s (A) and long 30–60 s (B) imaging intervals. **(C)** Quantification of Pearson's correlation coefficients between intercellular Mup1-GFP and either Sec7-mCherry (gray) or Vps4-mCherry (green) signal from steady state images acquired at indicated times during methionine pulse-chase (n = >27 cells), *, P < 0.0001 from unpaired Student's *t* test, individual values are included as jitter plot over histograms and represent n = 61–79 cells per condition. **(D)** Mander's overlap coefficient between Mup1-GFP and either Sec7-mCherry (gray) or Vps4-mCherry (green) from representative real-time methionine pulse-chase imaging experiment. **(E and F)** Time-lapse Airyscan micrographs of Mup1-GFP and Sec7-mCherry (E) or Vps4-mCherry (F) were analyzed by Zen Black colocalization software, and regions of Manders overlap (not signal colocalization) above background that were detected are depicted in white, with zoomed-in representations of the PM and endosome. Scale bar, 5 µm (white); 0.5 µm (yellow).

A). To explore this hypothesis more in detail, we created HA-tagged versions of WT (Ist1[WT]-HA) and a mutant resistant to ubiquitination (Ist1[KR]-HA), which has all 38 lysine residues mutated to arginine. Although levels of Ist1[KR]-HA are somewhat reduced (Fig. 7 A), a cycloheximide chase to inhibit translation reveals that it has similar stability to Ist1[WT]-HA over a 2-h chase

period (Fig. 7, B and C; and Fig. S4, A–C). Similarly, not only does the mutant Ist1[KR] tagged with GFP persist in cells, but it localizes to Vps4-positive endosomes that are distinct from Sec7 compartments (Fig. 7, D and E). Although Ist1[KR] mutants have similar behavior, only Ist1[WT]-HA supports the recycling defect Mup1-GFP expressed in *ist1Δ* cells, with Mup1-GFP retained at Vps4-

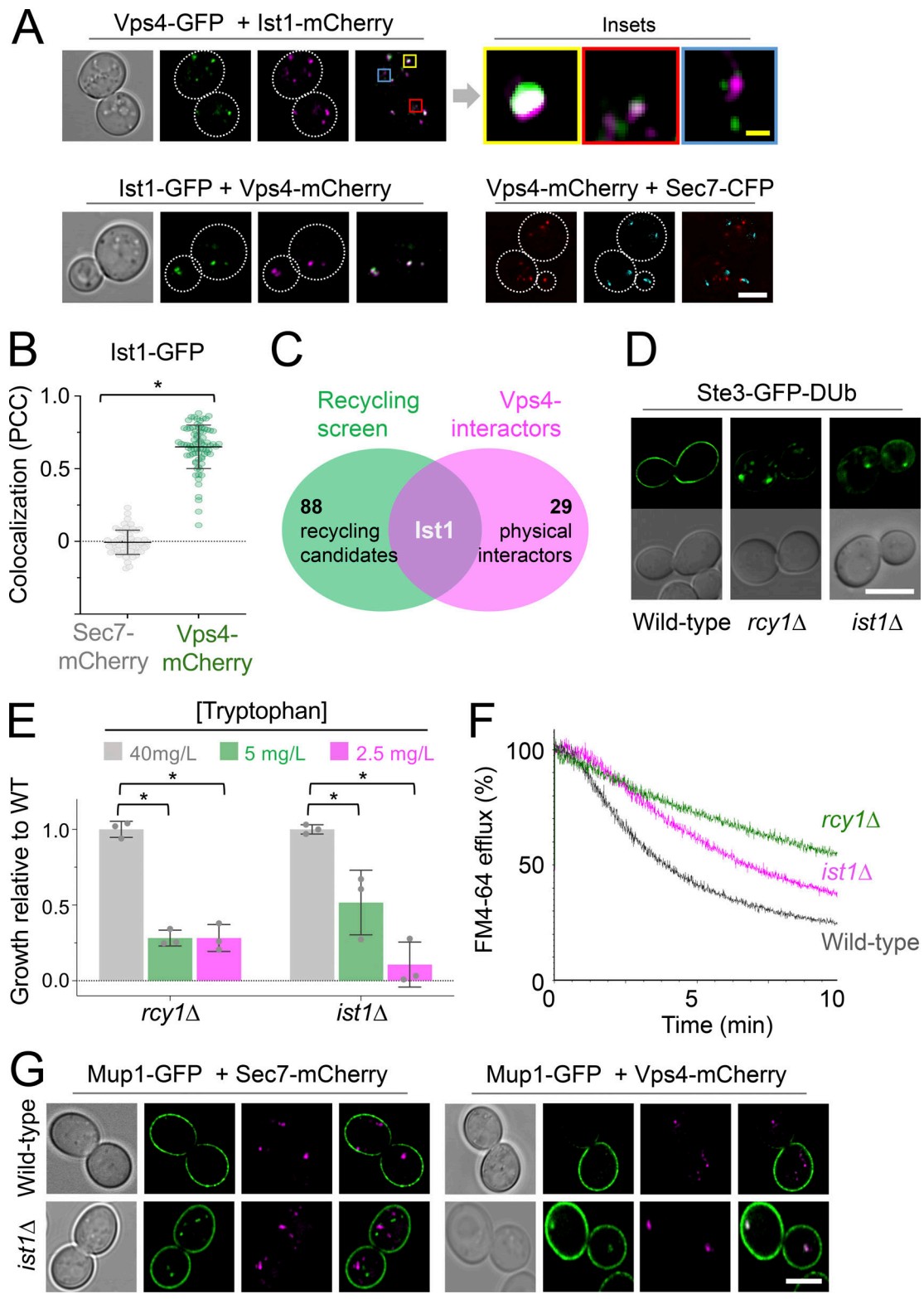

Figure 4. **Ist1 is required for endosomal recycling in yeast. (A)** WT cells coexpressing fluorescently labeled versions of Ist1, Vps4, and Sec7 were imaged by Airyscan microscopy. Insets show variation of colocalization. **(B)** Pearson's correlation coefficient's calculated For Ist1-GFP with Sec7-mCherry (gray) and Vps4-mCherry (green). *, P < 0.002 from unpaired *t* test; individual values are included as jitter plots over histograms and represent *n* = 27–72 cells per condition. **(C)** Venn diagram comparing 89 recycling factors (green) with known physical interactors of Vps4 (pink). **(D)** Localization of stably integrated Ste3-GFP-DUb in indicated strains by Airyscan microscopy. **(E)** Histogram showing relative growth of *rcy1Δ* and *ist1Δ* mutants compared with WT cells across media containing indicated concentrations of tryptophan. *, P < 0.02 from unpaired *t* test, *n* = 3. **(F)** FM4-64 efflux measurements from WT, *rcy1Δ*, and *ist1Δ* cells loaded with dye for 8 min at RT followed by three times ice-cold media washes. **(G)** Airyscan microscopy images of cells co-expressing Mup1-GFP and Sec7-mCherry (left) or Vps4-mCherry (right) in WT or *ist1Δ* cells. Scale bars, 5 μm (white); 0.5 μm (yellow).

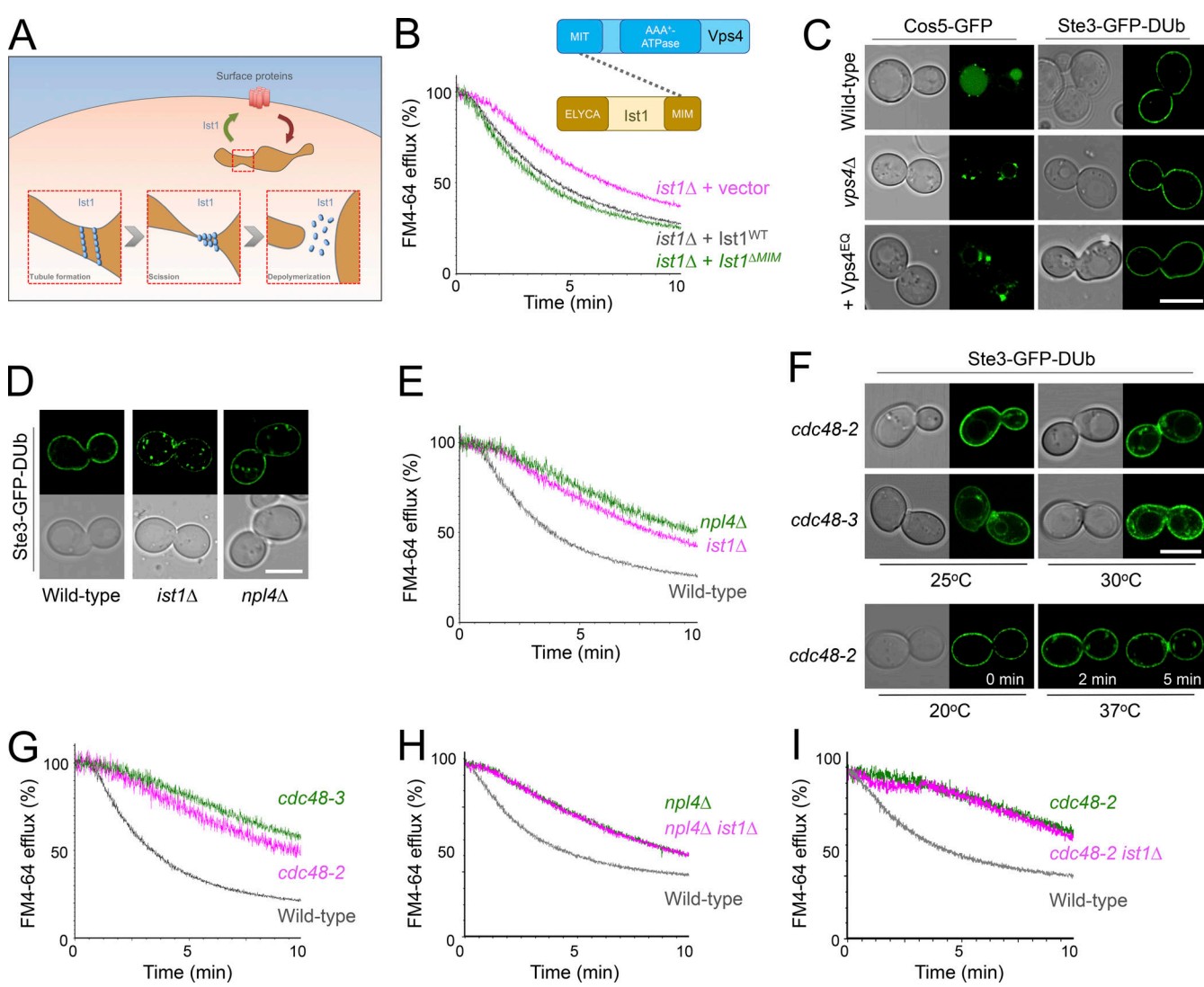

Figure 5. **Npl4-Cdc48 implicated in Ist1-mediated recycling pathway. (A)** Schematic representation of model for how yeast Ist1 could drive recycling, based on studies on human IST1 that drives polymerization/scission of endosome tubules to return material to the surface. **(B)** Efflux measurements were recorded from *ist1Δ* cells transformed with either vector control or plasmids expressing Ist1[WT] or Ist1[ΔMIM] loaded with FM4-64 for 8 min at RT followed by ice-cold washes. Cartoon representation of domain interaction between Vps4 and Ist1 included above. **(C)** Airyscan confocal microscopy of Cos5-GFP from a plasmid (left) or stably integrated Ste3-GFP-DUb (right) expressed in WT and *vps4Δ* cells, and also in the presence of Vps4[EQ] expressed from the *CUP1* promoter in the presence of 100 µM copper chloride. **(D)** Stably integrated Ste3-GFP-DUb expressed in WT, *ist1Δ*, and *npl4Δ* cells imaged by Airyscan confocal microscopy. **(E)** FM4-64 efflux measurements from indicated strains WT, *npl4Δ*, and *ist1Δ* cells grown to mid-log phase prior to loading with dye for 8 min at RT and efflux measured after washes. **(F)** Stably integrated Ste3-GFP-DUb was expressed in strains haboring temperature-sensitive alleles of *CDC48* (*cdc48-2* and *cdc48-3*), grown to mid-log phase at 25°C, and imaged by Airyscan confocal microscopy directly or following a 30-min incubation at 30°C. **(G–I)** Efflux measurements from indicated cells were first loaded with FM4-64 for 8 min, washed three times prior to cytometry. Scale bar, 5 µm.

endosomes in cells expressing Ist1[KR]-HA (Fig. 7 F). Furthermore, only Ist1[WT]-HA supports recycling of FM4-64 in *ist1Δ* cells to WT levels, but Ist1[KR]-HA cannot (Fig. 7 G).

Although our evidence implicates Ist1 ubiquitination in recycling, this approach to create a lysine-less version necessitated a large number (38/298) of point mutations. To confirm the stability of Ist1[KR] in vitro, we optimized expression of His[6] tagged versions of Ist1[WT] and Ist1[KR] in bacterial cells (Fig. S4 D). Recombinant Ist1[WT] and Ist1[KR] had similar yields and levels of purity (Fig. 7, H and I). Analyzing purified protein by nanoscale differential scanning fluorimetry (nanoDSF) detected aberrations in protein stability at similar stages of the heat ramp, with

Ist1[WT] at 35.3 ± 3.2°C and Ist1[KR] at 38.5 ± 1.4°C (Fig. 7, J and K). Furthermore, analysis of the scattering profiles, an indirect measurement of aggregation during heat-induced unfolding, showed no obvious indication that either Ist1[WT] or Ist1[KR] was unstable at physiological temperature (Fig. 7, L and M).

One caveat to the recycling defects observed above (Fig. 7) is that the Ist1[KR]-HA mutant levels are lower than WT, so it is unclear whether reduced cellular protein levels or lack of ubiquitination sites in Ist1[KR] are responsible for defects in recycling. To distinguish between these possibilities, we sought a system that would allow precise control of WT *IST1* expression, allowing it to be reduced to approximately that of the Ist1[KR]

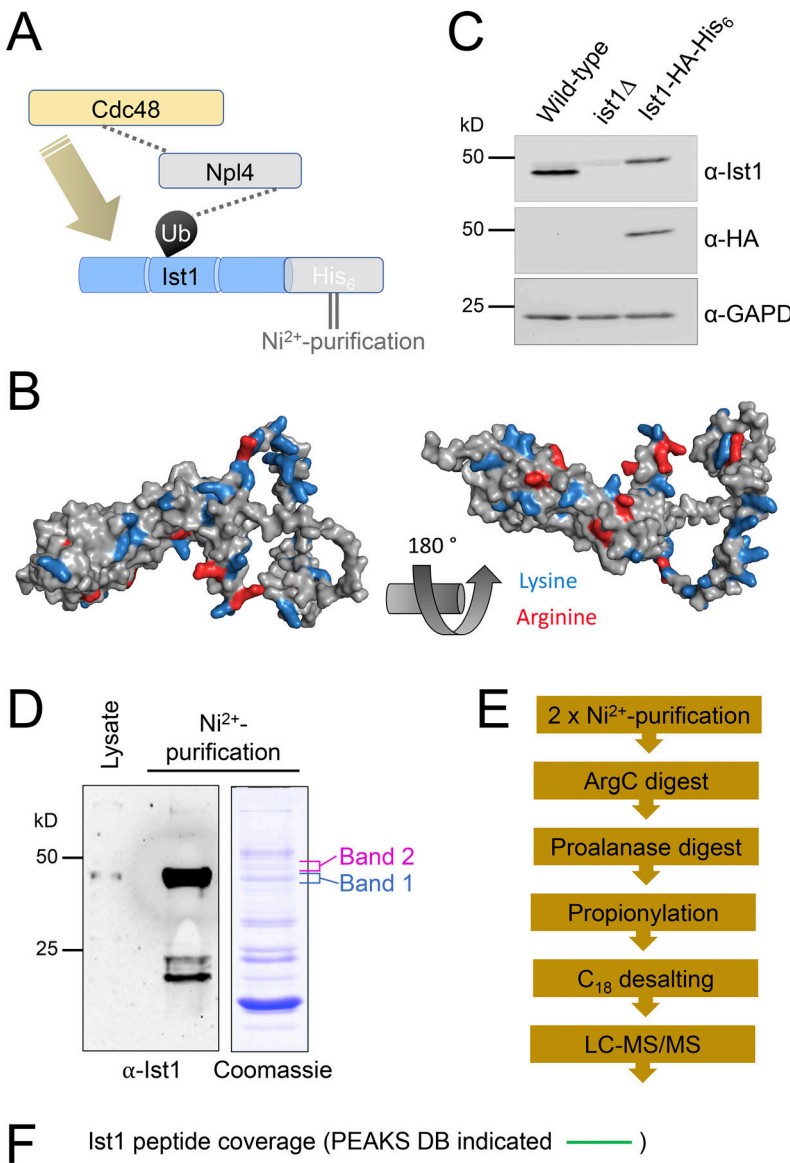

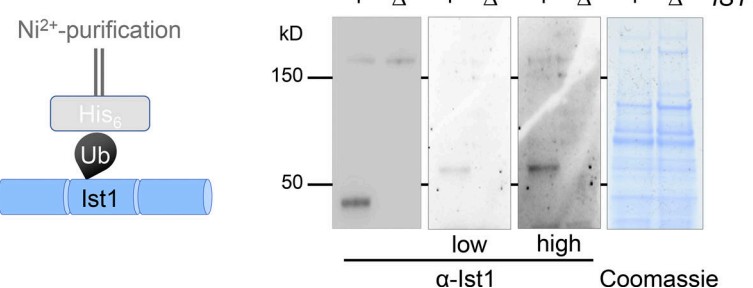

Figure 6. **Ist1 is ubiquitinated in vivo. (A)** Simplified representation of known interactions (dotted lines) between Cdc48, Npl4, and ubiquitin (Ub). We hypothesize this Npl4-Cdc48 enzyme module functionally connects with Ist1 via Ist1 ubiquitination. **(B)** Alphafold structural model of yeast Ist1 with lysine (blue) and arginine (red) residues indicated. **(C)** Immunoblot of lysates from WT, *ist1Δ*, and Ist1-HA-His$_6$ strains using α-Ist1, α-HA, and α-GAPDH antibodies. **(D)** Whole cell lysates were generated from cells expressing Ist1-HA-His$_6$ and run on the same SDS-PAGE gel as 0.5% of the purified elution from 2 liters culture followed by immunoblotting using α-Ist1 antibodies (left). A separate gel with 10% of purified sample was stained with Coomassie (right) and the bands excised for MS-based identification indicated. **(E)** Flow diagram depicting sequence of sample preparation for MS analysis of Ist1 targeted at identifying ubiquitinated peptides. **(F)** Ist1 amino acid sequence annotated with identified peptides (green) and potentially ubiquitinated lysine residue (pink) shown from PEAKS analysis. **(G)** Simplified schematic of Ni$^{2+}$-NTA purification of the ubiquitome to test is Ist1 is ubiquitinated. **(H)** Immunoblot using α-Ist1 antibodies of lysates generated from His$_6$-ubiquitin with (+) or lacking (Δ) *IST1* (left). 2 liters from each of these cells was purified via Ni$^{2+}$-NTA twice and analyzed by immunoblot and levels indicated by Coomassie staining of SDS-PAGE gels (right). Source data are available for this figure: SourceData F6.

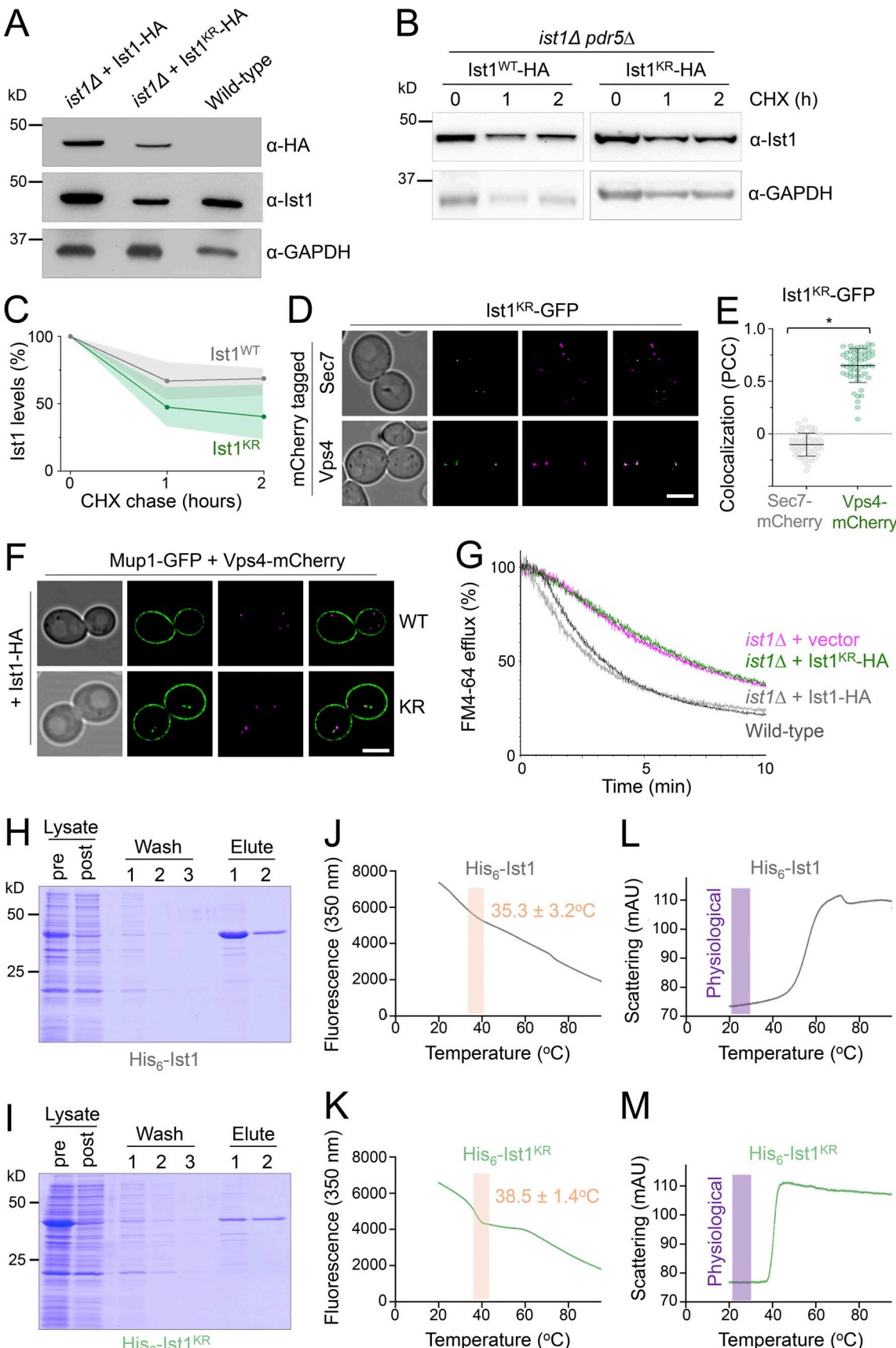

Figure 7. **Lysine-less Ist1 is defective in endosomal recycling. (A)** Immunoblot with indicated antibodies of lysates generated from *ist1Δ* cells transformed with plasmids expressing Ist1[WT]-HA or Ist1[KR]-HA from endogenous promoter, alongside WT cells. **(B)** Cycloheximide chase experiments using *ist1Δ pdr5Δ* cells

expressing Ist1[WT]-HA or Ist1[KR]-HA. Cells grown to mid-log phase were then exposed to 25 mg/liter cycloheximide for the denoted time before harvesting and immunoblot with α-Ist1 and α-GAPDH antibodies. **(C)** Graph of Ist1 stability following cycloheximide chase at indicated times of Ist1[WT] (gray) and Ist1[KR] (green), with SD (n = 3) indicated with respective shaded regions. **(D)** Airyscan images of *ist1Δ* cells co-expressing Ist1[KR]-GFP and Sec7-mCherry (upper) or Vps4-mCherry (lower) grown to mid-log phase. **(E)** Associated jitter plots from D showing Pearson's correlation coefficient. *, P < 0.0001 from unpaired t test, n = 56 cells per condition. **(F)** Airyscan microscopy of *ist1Δ* cells coexpressing Mup1-GFP, Vps4-mCherry with either Ist1[WT]-HA (upper) or Ist1[KR]-HA (lower). **(G)** FM4-64 efflux measurements from WT cells (dark gray) and *ist1Δ* mutants expressing plasmid borne copies of Ist1[WT]-HA (light gray), Ist1[KR]-HA (green) or transformed with an empty vector (pink). **(H and I)** Plasmids of His$_6$-Ist1[WT] and His$_6$-Ist1[KR] were expressed in BL21 DE3 codon optimized *E. coli* strain using 0.5 mM IPTG at 15°C overnight. Lysates were generated by sonication and bound to 600 μl Ni$^{2+}$-NTA bed volume, followed by washing in 20 mM imidazole and elution in 500 mM imidazole. Samples were analyzed by SDS-PAGE followed by Coomassie staining, showing protein levels in lysate pre- and postbinding to beads, the material lost during washes, and the final eluted products. **(J–M)** Intrinsic fluorescence at 350 nm (J and K) and unfolding induced aggregation via scattering in milli-absorbance units (mAU; L and M) was measured by nanoDSF for purified His$_6$-Ist1[WT] and His$_6$-Ist1[KR] samples exposed to a 1°C/min heat ramp. Scale bar, 5 μm. Source data are available for this figure: SourceData F7.

mutant. For this, we took advantage of a yeast estradiol with titratable induction (YETI) system (Arita et al., 2021) that allows *IST1* expression to be controlled with titration of β-estradiol/E2 (Fig. 8 A). Optimizing 6-h E2 treatments and assessing Ist1 levels by immunoblot revealed conditions that mimic: *ist1Δ* cells with no E2 in the media; Ist1[WT] with 0.7 ± 0.4 nM; and Ist1[KR] = 0.2 ± 0.04 nM (Fig. 8, B and C). We find that mimic of either Ist1[WT] or Ist1[KR] levels was sufficient to restore recycling of both Mup1-GFP and FM4-64 (Fig. 8, D and E).

In support of a model for Ubiquitin > Npl4 > Cdc48 regulating endosomal cycling of Ist1, we find elevated endosomal retention of Ist1[KR]-GFP, which cannot be ubiquitinated (Fig. S5, A and B). Furthermore, elevated endosomal levels of Ist1-GFP are observed in *npl4Δ* cells and mutants with reduced Cdc48 ATPase activity. Combining the lysine-less version of Ist1[KR]-GFP with Npl4/Cdc48 mutants results in its mislocalization to the nucleus (Fig. S5, C and D), which precluded assessment of endosomal recruitment but further alludes to a biological connection between ubiquitinated Ist1 and Npl4-Cdc48. To better implicate Ist1 ubiquitination in recycling, we generated a DUb-fusion of Ist1-GFP and a catalytically inactive dub[C>S] counterpart (Fig. 9 A). As expected, neither Ist1-GFP-DUb nor Ist1-GFP-dub[C>S] localizes with Sec7 compartments (Fig. 9, B and E). In agreement with Ist1[KR] experiments, we find that Ist1-GFP-DUb is defective in recycling to the same degree as *ist1Δ* cells (Fig. 9 F). Introducing only a single point mutation to ablate catalytic activity on the enzyme (Ist1-GFP-dub[C>S]) is sufficient for the fusion protein to recycle FM4-64 to WT levels (Fig. 9 G). Although perturbing Ist1-GFP ubiquitination by fusion with a DUb results in colocalization with the large, static prevacuolar Vps4 population, there is an observable separation of Vps4 and Ist1 in the small mobile population (Fig. 9, C–E). This suggests that although Vps4 and Ist1 occupy the same compartments in WT conditions, the distinct trafficking intermediates can emanate from these endosomes to recycle material back to the surface.

## Discussion

We show that retrograde recycling of Snc1/Snc2 is distinct from the recycling of the Mup1/Fur4 nutrient transporters. Snc1/2 mainly internalizes to Sec7-marked TGN compartments, exhibits polarized distribution in daughter cells, and relies on cargo ubiquitination (Fig. 1). In contrast, Mup1/Fur4 internalizes to Vps4-marked endosome compartments, localizes

predominantly to mother cells during budding, and deubiquitination triggers their recycling. As Snc1/2 is the R-SNARE carried on secretory vesicles that feed material into the growing daughter cell, it would be expected to transit the Golgi. However, nutrient transporters may not provide physiological benefits to burgeoning daughter cells as their surface residence is maintained by recycling in only mother cells.

Ubiquitination of nutrient transporters like Mup1 in response to substrates (Schothorst et al., 2013) can be counteracted by substrate removal, which triggers cargo recycling from Vps4 compartments back to the PM (Figs. 2 and 3). We note that the small levels of colocalization detected between Mup1-GFP and Sec7-mCherry were only observed when Sec7 signal was adjacent to the PM (Fig. 3 F). This marginal level of colocalization with peripheral Golgi compartments may have derived from PM-localized Mup1-GFP following deconvolution, and not an internalized endosome population. Given that Mup1 internalizes to Vps4 endosomes on the scale of seconds to minutes, these compartments can be considered yeast early endosomes. Furthermore, our interpretation of the imaging and biochemical data presented is internalized Mup1 recycles back to the surface directly from these endosome compartments, allowing them to be considered yeast recycling endosomes. It is tempting to speculate that the large static endosome population of Vps4, which is also termed the prevacuolar compartment due to its vicinity to the vacuole, represents functionally definable MVBs. Other Vps4 and Ist1 positive compartments might be more akin to sorting or recycling endosomes, which have not matured or committed to ILV formation and surface protein degradation. Although endosomal recycling of cargo in mammalian cells is complex (Grant and Donaldson, 2009; Goldenring, 2015; O'Sullivan and Lindsay, 2020), these observations support the notion that recycling features are evolutionarily conserved and can be elucidated using yeast.

To this end, we reveal mechanisms of action of the yeast ortholog of mammalian IST1, which regulates endosomal recycling in animal cells (McCullough et al., 2015; Allison et al., 2013, 2017). We demonstrate that yeast Ist1 is required for recycling proteins and lipids back to the PM (Fig. 4). We find Ist1 is ubiquitinated (Fig. 6) and a lysine-less Ist1[KR] is not functional in endosomal recycling due to its lack of ubiquitin sites (Figs. 7 and 8). Beyond this, we used a ubiquitination reversal strategy that also demonstrated that Ist1 ubiquitination is required for recycling (Fig. 9). Recently, regulation of the IST1 partner CHMP1B

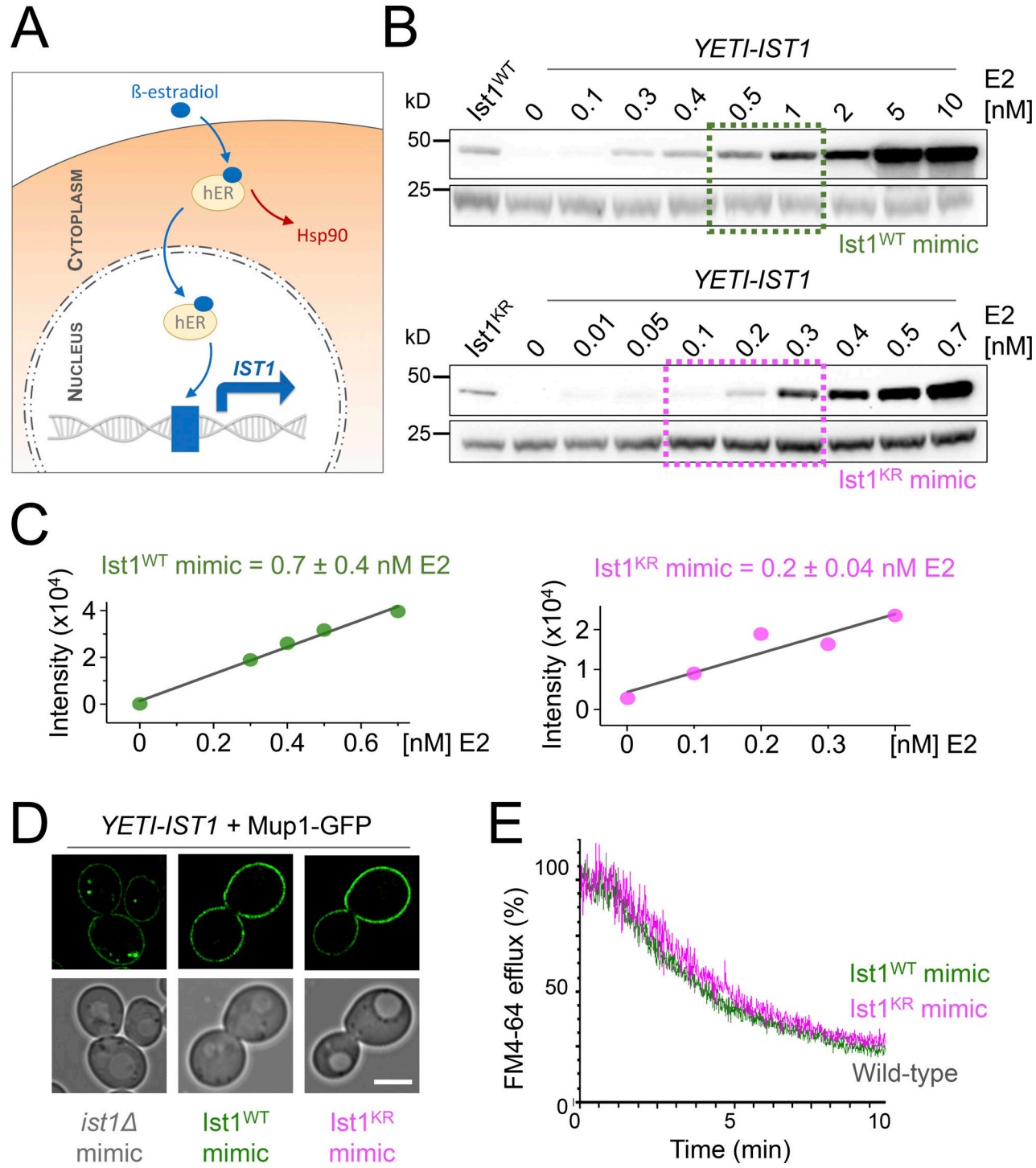

**Figure 8. Reduced levels of ubiquitinatable Ist1 are sufficient for recycling. (A)** Schematic showing YETI system to regulate levels of *IST1* expression. **(B and C)** Immunoblot depicting Ist1 levels from *YETI-IST1* cells exposed to indicated β-estradiol titrations with concentrations used to mimic Ist1[WT] (green) and Ist1[KR] (pink) levels estimated with SD (*n* = 3) shown. **(D)** Airyscan microscopy of Mup1-GFP expressed in *YETI-IST1* cells exposed to β-estradiol concentrations required to mimic *ist1Δ*, Ist1[WT], and Ist1[KR] levels for 6 h prior to imaging. **(E)** FM4-64 efflux measurements from WT cells and *YETI-IST1* cells exposed to β-estradiol to mimic Ist1[WT] and Ist1[KR] levels prior to loading with dye for 8 min at RT. Scale bar, 5 μm. Source data are available for this figure: SourceData F8.

has also been shown through ubiquitination (Crespo-Yàñez et al., 2018), alluding to general posttranslational means to control ESCRT-polymerization in other modes of actions. Super-resolution imaging, especially combined with the DUb fusion strategy to inhibit Ist1-mediated recycling, shows that non-Golgi

Ist1-endosomes can be observed ± Vps4, further alluding to complexity of endosomal organization in yeast. Whilst it was expected that Sec7 would not localize to these recycling endosomes, it is curious that Ist1 compartments labeled by the non-functional Ist1-GFP-DUb, but lacking Vps4, are more obvious

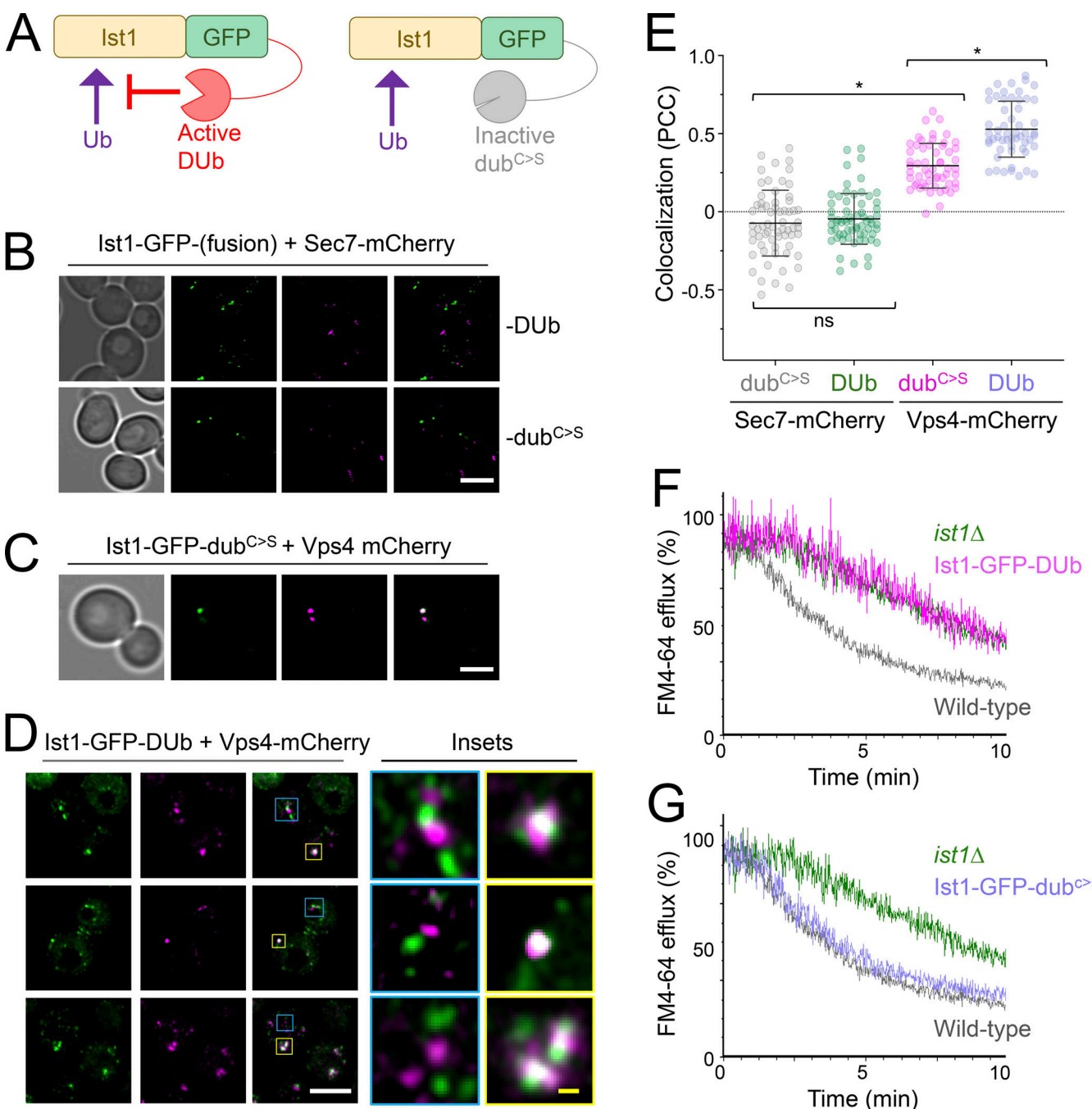

Figure 9. **Reversal of Ist1 ubiquitination inhibits endosomal recycling. (A)** Schematic illustrating the strategy to fuse Ist1-GFP to an active DUb and an inactive catalytically dead (dub$^{C>S}$) mutant enzyme. **(B–D)** Airyscan microscopy of indicated fluorescent proteins expressed in WT cells grown to exponential phase prior to imaging. Insets (D) show of colocalization and distinct foci in yellow and blue, respectively. **(E)** Jitter plots showing Pearson's correlation coefficient (PCC) from imaging shown in B–D. *, P < 0.0001 from unpaired *t* test, *n* = 55–65 cells per condition. **(F and G)** FM4-64 efflux measurements from dye loaded to cells expressing active Ist1-GFP-DUb (F) and inactive Ist1-GFP-DUb$^{C>S}$ (G) fusion proteins, with profiles from WT and *ist1*Δ mutants included for reference. Scale bar, 5 µm (white); 0.5 µm (yellow).

and numerous. The residence of Vps4 and Ist1 on overlapping endosome populations might allow fine-tuning of the surface proteome, for example, in response to metabolic demand. If cargo is destined for recycling from endosomes, the role of Ist1 is prioritized, but if sufficient ubiquitinated cargoes accumulate, Vps4 and ESCRT coalescence (Banjade et al., 2022) predominates to drive cargo degradation.

In addition to ubiquitination of Ist1, we show the ubiquitin-binding adaptor Npl4 and its ATPase partner Cdc48 (p97/VCP) regulate endosomal recruitment of Ist1 and are required for efficient cargo recycling to the PM (Fig. 5). It may be that the role of Ist1 in yeast recycling depends on its polymerization, analogous to its mammalian counterpart. The most likely functional connection to Ist1 ubiquitination being required for recycling is

via the ubiquitin-binding protein Npl4 and Cdc48, especially as combining these mutations does not exacerbate recycling defects (Fig. 5, H and I). This speculative model collectively unites many trafficking observations in different systems and argues conserved mechanisms drive distinct yeast recycling pathways.

## Materials and methods

### Reagents
Supplemental tables are included to document use of plasmids (Table S1), yeast strains (Table S2), and primary antibodies (Table S3).

### Cell culture
*S. cerevisiae* yeast strains were cultured in synthetic complete (SC) minimal media (2% glucose, yeast nitrogen base supplemented with base/amino acid drop-out mixtures for selections) or yeast extract peptone dextrose (YPD; 2% glucose, 2% peptone, and 1% yeast extract). Yeast cultures were typically grown in serial dilution overnight to allow for harvesting at mid-log phase ($OD_{600} \leq 1.0$) prior to experimentation unless otherwise stated. Selection of strains harboring KanMX cassettes was carried out in rich media containing 250 µg/ml geneticin/G418 (Formedium). The mCherry and GFP fusions of *SEC7* were generated with a methotrexate cassette selected on SC media containing 20 mM methotrexate (Alfa Aesar) and 5 mg/ml sulfanilamide before the *loxP* flanked cassette was excised by transient expression from a *TEF1-Cre* plasmid that was subsequently removed via 5-fluoroorotic acid media (MacDonald and Piper, 2015). Expression from the *CUP1* promoter was induced by the addition of 20–100 µM copper chloride. Methionine (20 µg/ml) and uracil (40 µg/ml) were added to SC media to induce trafficking of Mup1 and Fur4, respectively. Cycloheximide chase experiments were performed using yeast harboring a *pdr5Δ* mutation grown to mid-log phase in SC media followed by growth in media containing 25 mg/l cycloheximide form indicated time points prior to lysate generation. YETI control of *IST1* was achieved by addition of indicated concentrations of β-estradiol to SC growth media for 6 h.

### Confocal microscopy
Yeast cells were harvested from mid-log phase ($OD_{600} \leq 1.0$) and prepared for imaging by concentrating in SC media. Microscopy was performed with Zeiss laser scanning confocal instruments (LSM780/LSM710 for standard confocal or LSM880 equipped with an Airyscan and LSM980 equipped with Airyscan2 detectors) using Plan-Apochromat 63×/1.4 objective lenses. The fluorescent proteins mCherry, photo-converted mEOS, and mStrawberry were excited using the 561-nm line from a yellow diode-pumped solid-state laser, and the emission range 570–620 nm was collected. The fluorescent proteins mGFP, GFP, and mNeonGreen, and preconverted mEOS were excited using the 488-nm line from an Argon laser, and the emission range 495–550 nm was collected. The fluorescent protein mEOS was photo converted using 0.5% of the 405-nm laser with five iterations per conversion and three conversions of a defined region of interest as stated. YPD containing 0.8 µM FM4-64 was used to label vacuoles for 1 h followed by three times washing and 1-h chase period in SC minimal media. For dual population imaging, cultures were grown independently to the mid-log phase before mixing 1:1 ratio and grown for a further 1–3 h. 5 µg/ml Hoescht was added to harvested cells for 10 min prior to imaging, the cells were excited using the 405-nm line, and the emission 460/50 nm was collected.

### Microfluidics and time-lapse microscopy
Yeast cultures were grown to very early log phase ($OD_{600} \leq 0.2$) and adhered to 35-mm glass-bottom coverslip dishes (Ibidi GmbH) coated with concanavalin A (Sigma-Aldrich) prior to live-cell imaging at RT in appropriate SC media. Concanavalin A coating was prepared by adding 1 mg/ml concanavalin A in water to the glass-bottom coverslip for 5 min prior to three washing steps; prepared plates were routinely stored at 4°C. Sterile media exchanges were performed using 50-ml syringes through tubing fused to the lid of the 35-mm dishes.

### Image analysis
Airyscan micrographs were processed using Zen Blue or Zen Black software (Zeiss) and were further modified using Fiji. For time-lapse movies, bleach correction was carried out using the inbuilt Fiji plugin and histogram-matching method (Miura, 2020). Any necessary drift correction was carried out in Fiji using the plugins Hyper Stack Reg and Turbo Reg (Thevenaz et al., 1998). Fluorescence intensity measurements during photoconversion experiments were assessed in Zen Black. Steady state colocalization measurements of cells expressing Mup1-GFP in Fig. 3 C were performed using cell magic wand and morphological erosion to exclude surface signal prior to Pearson correlation coefficients being calculated. For other colocalization analyses, as indicated, Mander's overlap coefficients or Pearson's correlation coefficients were calculated using Zen Blue/Zen Black (Zeiss) following normalization for cells expressing individual GFP and mCherry fluorescent proteins. Immunoblot intensities were measured using ImageJ and normalized to background and GAPDH signal. To estimate percentage endosome GFP signal, the intensity was compared from endosomes segmented using otsu thresholding and from the whole cell. All data were then plotted in GraphPad (v9.0.2, Prism).

### FM4-64 recycling assay
Yeast cultures were grown to mid-log phase in SC minimal media with corresponding selection to plasmid or YPD, 1 ml of cells (OD = 1.0) were harvested, incubated for 8 min at RT in 100 µl YPD containing 40 µM FM4-64 dye (*N*-[3-Triethylammoniumpropyl]-4-[6-[4-[Diethylamino] Phenyl] Hexatrienyl] Pyridinium Dibromide) dye. Labeled cells were then washed in ice-cold SC media for 3 min on ice, three times. Final wash concentrated cells in 100 µl SC media for preparation for flow cytometry. Approximately 2,500 cells flowed per second at ~600 V using LSR Fortessa (BD Biosciences), over a 10-min time period, and the FM4-64 intensity was measured with excitation at 561 nm, laser filter 710/50. Background autofluorescence was recorded using measurements from the 530/50 nm detector.

## Immunoblotting

Equivalent amounts of yeast culture grown to mid-log phase ($OD_{600}$ = <1.0) were harvested, treated with 500 µl 0.2 N NaOH for 3 min, and then resuspended in lysis buffer (8 M urea, 10% glycerol, 5% SDS, 10% 2-mercaptoethanol, 50 mM Tris HCl, pH 6.8, 0.1% bromophenol blue). Proteins were resolved using SDS-PAGE and transferred to nitrocellulose membrane using the iBlot2 transfer system (Thermo Fisher Scientific). The membrane was probed using labeled antibodies and visualized using super signal Pico Plus (Thermo Fisher Scientific). Enhanced chemiluminescence signal intensity was captured using an iBright Imager (Thermo Fisher Scientific).

## Statistical tests

Indicated statistical tests for experimental comparisons were performed using GraphPad (v9.0.2, Prism). An asterisk is used in graphs to denote statistically significant differences. Statistical tests used throughout the manuscript and P values are shown in Table S4.

## Bioinformatics

Gene ontology term finder (Cherry et al., 2012) was used to analyze all the results of the genetic screen for recycling machinery described in MacDonald and Piper (2017), with searches for specific enzyme activity shown. The physical interactome was acquired from YeastMine (Balakrishnan et al., 2012).

## Tat2 recycling assay

Tryptophan auxotroph (*trp1Δ*) yeast cells based on the SEY6210 background were grown to mid-log phase before being spotted out across a 10-fold serial dilution and grown on plates of replete (40 mg/liter) and two restricted (5 and 2.5 mg/liter) tryptophan concentrations. To quantify growth, densitometry was used to measure the growth intensity across different dilutions on the plate (Paine et al., 2021 Preprint). Yeast growth at each dilution was normalized to a WT control on the same plate, and the difference was plotted for indicated tryptophan concentrations.

## Protein purification from yeast

Yeast CMY158 and CMY2056 optimized for ubiquitin purification (MacDonald et al., 2017; MacDonald et al., 2020) expressing $His_6$-tagged ubiquitin, or yeast CMY2042 expressing Ist1-HA-$His_6$ were grown in a 2-liter culture to mid-log phase before they were harvested and treated with 0.2 M NaOH for 3 min and brought up in denaturing lysis buffer (8 M urea, 20 mM sodium phosphate, pH 7.4, 300 mM NaCl, 2.5% SDS, and 5 mM 2-mercaptoethanol). The lysates were diluted 80-fold using dilution buffer (8 M urea, 20 mM sodium phosphate, pH 7.4, 300 mM NaCl, 5 mM 2-mercaptoethanol) and bound to a 2 ml bed of $Ni^{2+}$-NTA agarose beads for 2 h at RT. Beads were incubated with wash buffer (8 M urea, 20 mM sodium phosphate, pH 7.4, 300 mM NaC, 5 mM 2-mercaptoethanol, and 5 mM imidazole) five times before eluting using wash buffer at pH 4.5. The sample was then neutralized and bound to 100 µl bed of $Ni^{2+}$-NTA agarose beads for 2 h before repeat washes and elution in dilution buffer containing 500 mM imidazole. Loading buffer was added to the samples for downstream SDS-PAGE analysis.

## Recombinant protein expression and purification

Protein expression for $His_6$-$Ist1^{WT}$ and $His_6$-$Ist1^{KR}$ was performed using BL21-DE3 (Invitrogen) and BL21-CodonPlus (Agilent) *Escherichia coli* strains grown in 1 liter of LB broth to $OD_{600}$ = 0.6 before induction with 0.5 mM IPTG for 16 h at 15°C. The cell pellet was resuspended in buffer A (20 mM sodium phosphate, pH 7.4, 300 mM NaCl, and 30 mM imidazole) containing cOmplete protease inhibitor cocktail (Roche) and sonicated, and the lysate was clarified at 18,000× *g* and bound to 600 µl bed of $Ni^{2+}$-NTA agarose beads for 2 h at 4°C. Beads were washed five times using 10 ml buffer A before elution with buffer B (20 mM sodium phosphate, pH 7.4, 300 mM NaCl, and 500 mM imidazole). Protein concentrations assays were performed using Pierce BCA Protein Assay Kit (Thermo Fisher Scientific), incubated at 37°C for 30 min before OD was measured at 560 nm.

## DSF

Prometheus NT.48 instrument (NanoTemper Technologies) performed DSF to determine $T_m$ and $T_{onset}$ of loaded protein samples. Purified protein samples in 20 mM sodium phosphate, pH 7.4, 300 mM NaCl, 500 mM imidazole were loaded in nanoDSF grade standard capillaries (NanoTemper Technologies) and laser power optimized for signal. Thermal stress from 20 to 95°C was performed on the sample shifting at a rate of 1°C/min. Fluorescence emission from the single Ist1 tryptophan (W134) after excitation with UV at 280 nm was collected at 330–350 nm. Aggregation of protein was assessed concurrently with back reflection optics. Thermal stability parameters $T_m$ and $T_{onset}$ were unbiasedly assessed and calculated by PR. ThermControl software (V.2.1.3).

## MALDI MS

Proteins were digested from gels with 0.02 µg/µl modified porcine trypsin (Promega) after reduction with dithioerythritol and iodoacetamide, washing two times with 50% (vol:vol) aqueous acetonitrile containing 25 mM ammonium bicarbonate, then once with acetonitrile, and dried in a vacuum concentrator for 20 min. Trypsin digests were incubated at 37°C before 1 µl of peptide mixture was applied directly to the ground steel MALDI target plate, followed immediately by an equal volume of freshly-prepared 5 mg/ml solution of 4-hydroxy-α-cyano-cinnamic acid (Sigma-Aldrich) in 50% aqueous (vol:vol) acetonitrile containing 0.1%, trifluoroacetic acid (vol:vol). Positive-ion MALDI mass spectra were acquired over a mass range of m/z 800–5,000 using a Bruker ultraflex III in reflectron mode, equipped with a Nd:YAG smart beam laser. Spectral processing and peak list generation for MS and MS/MS spectra were performed with Bruker flexAnalysis software (version 3.3). Tandem mass spectral data were searched with Mascot (Matrix Science Ltd., version 2.6.1), through the Bruker BioTools interface (version 3.2), against the *S. cerevisiae* subset of SwissProt. Peptide spectral matches were filtered to require an expect score of 0.05 or better.

## LC-MS/MS

Denatured protein was diluted to 2 M urea with aqueous 50 mM ammonium bicarbonate, reduced with 0.7 mg/ml dithiothreitol,

and heated at 55°C, and then alkylated with 1.9 mg/ml chloro-acetic acid before proteolytic digestion with 0.2 µg sequencing grade ArgC (Promega) and incubated at 37°C for 16 h. Peptide solution was then acidified with 0.1% formic acid before adding 0.2 µg Mass Spec grade ProAlanase protease (Promega) and then incubated at 37°C for a further 2 h. Resulting peptides were desalted using $C_{18}$ ZipTip filters (Millipore), eluted with 50% acetonitrile, and dried under vacuum. Half the peptide mixture was adjusted to pH 10 with ammonium hydroxide, and 10 µl of propionic anhydride was added and sample incubated at 60°C for 1 h. The other peptide mixture was not propionylated. Both sets of peptides were acidified with 0.1% trifluoroacetic acid, vacuum dried, and redissolved in aqueous 0.1% (vol:vol) tri-fluoroacetic acid followed by analysis by LC-MS/MS. Peptides were loaded onto a mClass nanoflow UPLC system (Waters) equipped with a nanoEaze M/Z Symmetry 100 Å $C_{18}$, 5 µm trap column (180 µm × 20 mm, Waters), and a PepMap, 2 µm, 100 Å, $C_{18}$ EasyNano nanocapillary column (75 µm × 500 mm, Thermo Fisher Scientific).

Separation used gradient elution of two solvents: solvent A, aqueous 0.1% (vol:vol) formic acid; solvent B, acetonitrile con-taining 0.1% (vol:vol) formic acid. The linear multistep gradient profile was as follows: 3–10% B over 7 min, 10–35% B over 30 min, 35–99% B over 5 min, and then proceeded for wash with 99% solvent B for 4 min. The nanoLC system was interfaced with an Orbitrap Fusion Tribrid mass spectrometer (Thermo Fisher Scientific) with an EasyNano ionization source (Thermo Fisher Scientific). Positive electrospray ionization–MS and $MS^2$ spectra were acquired using Xcalibur software (version 4.0, Thermo Fisher Scientific). Instrument source settings were as follows: ion spray voltage, 1,900 V; sweep gas, 0 Arb; and ion transfer tube temperature, 275°C. $MS^1$ spectra were acquired in the Or-bitrap with: 120,000 resolution, scan range: m/z 375–1,500; AGC target, $4e^5$; max fill time, 100 ms. Data-dependent acquisition was performed in top speed mode using a 1-s cycle, selecting the most intense precursors with charge states >1. Dynamic exclu-sion was performed for 50 s after precursor selection, and a minimum threshold for fragmentation was set at $5e^3$. $MS^2$ spectra were acquired in the linear ion trap with scan rate, turbo; quadrupole isolation, 1.6 m/z; activation type, HCD; ac-tivation energy: 32%; AGC target, $5e^3$; first mass, 110 m/z; and max fill time, 100 ms. Acquisitions were arranged by Xcalibur to inject ions for all available parallelizable time.

Peak lists were analyzed using PEAKS StudioX Pro (Bion-formatic Solutions, Inc.) and Byonic (Protein Metrics) against the *S. cerevisiae* subset of SwissProt. Peptide identifications were adjusted to achieve a 5% FDR as assessed empirically against a reverse database search.

### Supplemental material
Fig. S1 shows differential trafficking itineraries of SNAREs and nutrient transporters. Fig. S2 shows the role of Ist1 mutants and implication of Npl4 in endosomal recycling. Fig. S3 shows MS and pull-down controls to show Ist1 is ubiquitinated. Fig. S4 shows assessments of Ist1 stability. Fig. S5 shows Ist1 local-izations in vivo. Video 1 shows Mup1-GFP + Vps4-mCherry me-thionine pulse (short imaging intervals). Video 2 shows Mup1-GFP +

Sec7-mCherry methionine pulse (short imaging intervals). Video 3 shows Mup1-GFP + Vps4-mCherry methionine pulse-chase (long imaging intervals). Video 4 shows Mup1-GFP + Sec7-mCherry me-thionine pulse-chase (long imaging intervals). Video 5 shows Mup1-mEos photoconversion methionine pulse-chase (long imaging intervals). Video 6 shows Ist1-mCherry + Vps4-GFP time-lapse imaging. Video 7 shows Ist1-mCherry + Mup1-GFP time-lapse imaging. Table S1 shows yeast strains used in this study. Table S2 shows plasmids used in this study. Table S3 lists primary antibodies. Table S4 shows statistical analyses. Table S5 shows peptides identified by MS.

## Acknowledgments
We would like to thank Dave Katzmann (Mayo Clinic, Rochester, MN) for Ist1 antibodies and expression plasmids; David Teis (University of Innsbruck, Innsbruck, Austria) and Jeff Brodsky (University of Pittsburgh, Pittsburgh, PA) for yeast strains; Rob Piper (University of Iowa), Markus Babst (University of Utah), and Chris Stefan (Laboratory for Molecular Cell Biology, Uni-versity College London) for helpful discussions. Thanks also to the staff at York Bioscience Technology Facility, in particular Pete O'Toole, Jo Marrison, Karen Hogg, and Graeme Park, for help with imaging and photoconversion. Thanks to Jared Cart-wright and Rebecca Preece (Protein Production Laboratory) for advice on protein purification, and Andrew Leech (Molecular Interactions Laboratory) for nanoDSF training. Thanks also to Adam Dowle and Chris Powell (Centre of Excellence in Mass Spectrometry) for help with proteomics. Finally, thanks to Sarah Lecinski for advice on data analysis and Katherine Paine for comments on the manuscript.

This research was supported by a Sir Henry Dale Research Fellowship from the Wellcome Trust and the Royal Society 204636/Z/16/Z (C. MacDonald).

The authors declare no competing financial interests.

Author contributions: K.M.E. Laidlaw: methodology, valida-tion, formal analysis, investigation, writing—review and edit-ing, visualization. G. Calder: methodology, formal analysis, writing—review and editing, visualization. C. MacDonald: con-ceptualization, methodology, validation, formal analysis, inves-tigation, writing—original draft, writing—review and editing, visualization, supervision, project administration, funding acquisition.

Submitted: 28 September 2021

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

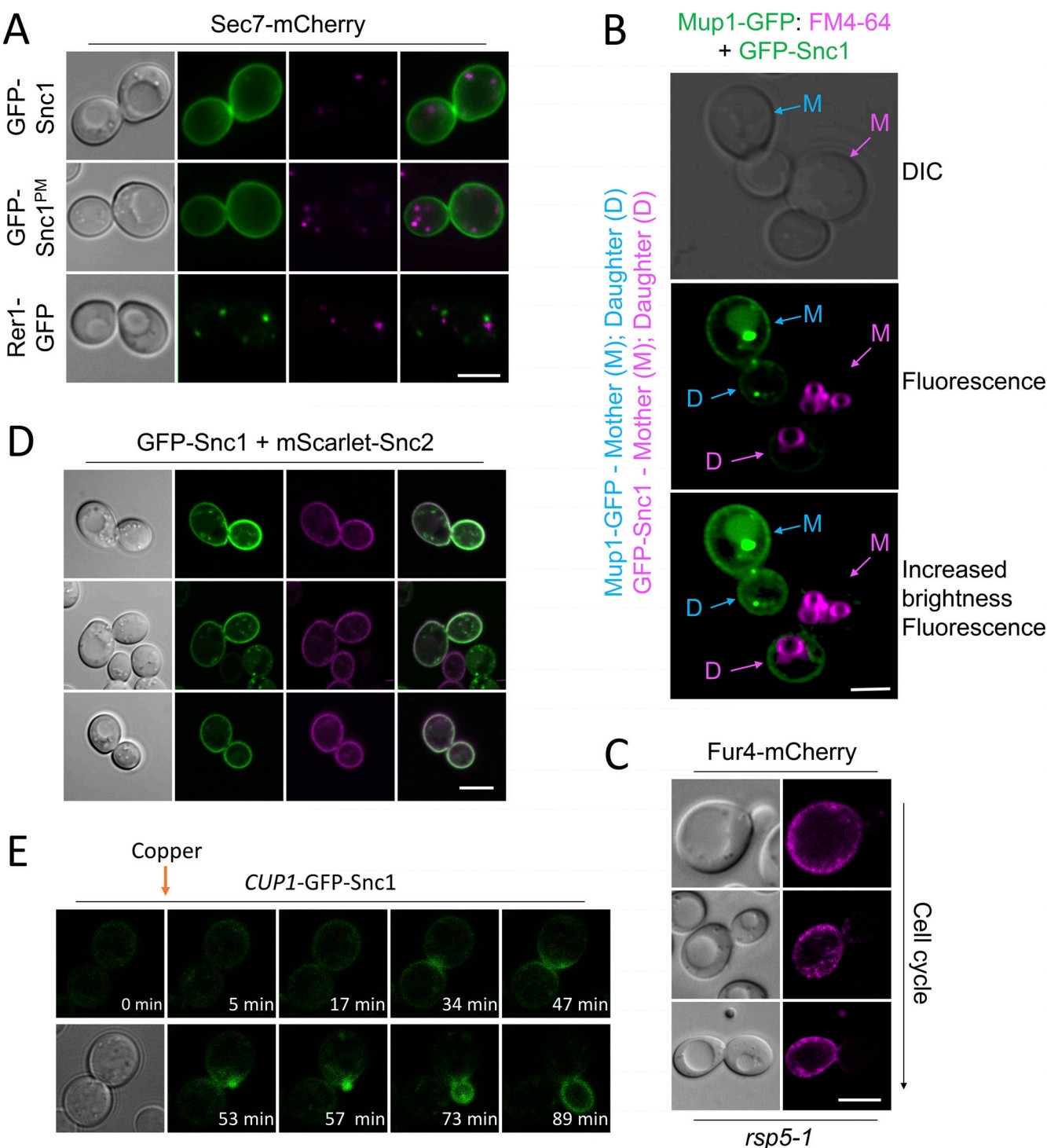

Figure S1. **Differential trafficking itineraries of SNAREs and nutrient transporters. (A)** Airyscan confocal microscopy of WT cells expressing an endogenously expressed Sec7-Cherry and indicated GFP-tagged proteins plasmids show expected localizations. **(B)** WT cells expressing GFP-Snc1 were mixed with Mup1-GFP expressing cells previously pulse-chased with FM4-64 prior to imaging. **(C)** *rsp5-1* cells expressing Fur4-mCherry were grown to mid-log phase in SC media and processed for confocal microscopy imaging. **(D)** Airyscan microscopy shows colocalization and correct localization of tagged versions of Snc1 and Snc2 expressed from the *CUP1* promoter. **(E)** Expression of GFP-Snc1 following microfluidic addition of 20 µM copper chloride to cells during time-lapse microscopy. Scale bar, 5 µM.

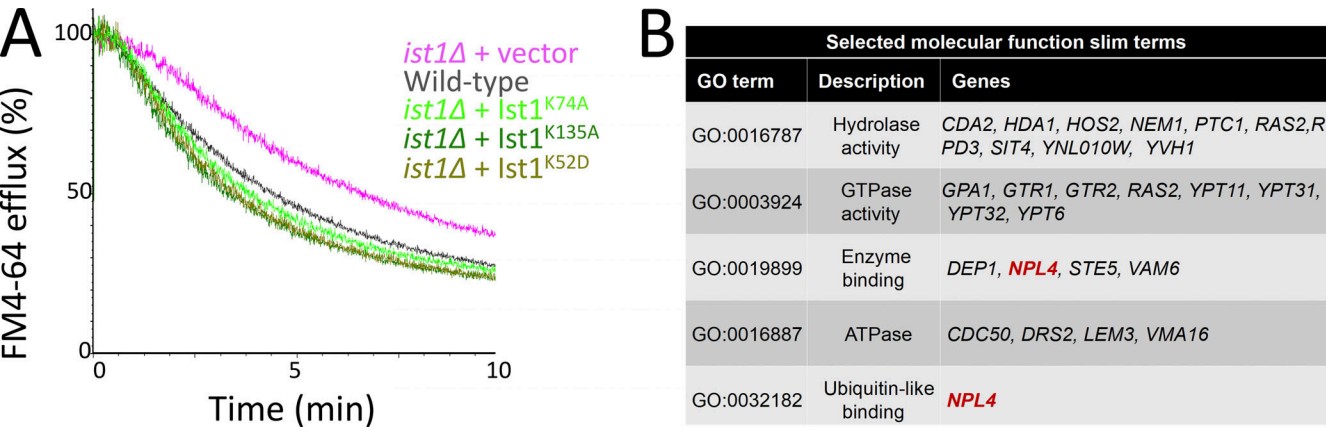

Figure S2. **The role of Ist1 mutants, and implication of Npl4, in endosomal recycling. (A)** Either WT cells or *ist1Δ* mutants transformed with vector (pink) or indicated Ist1 point mutations (green) were grown to mid-log phase prior to loading with dye for 8 min at RT, washing in cold media, and efflux measured by flow cytometry. **(B)** Table showing selected slim term annotations for recycling machinery associated with relevant enzyme activity that exhibit enrichment compared with genome-wide distribution.

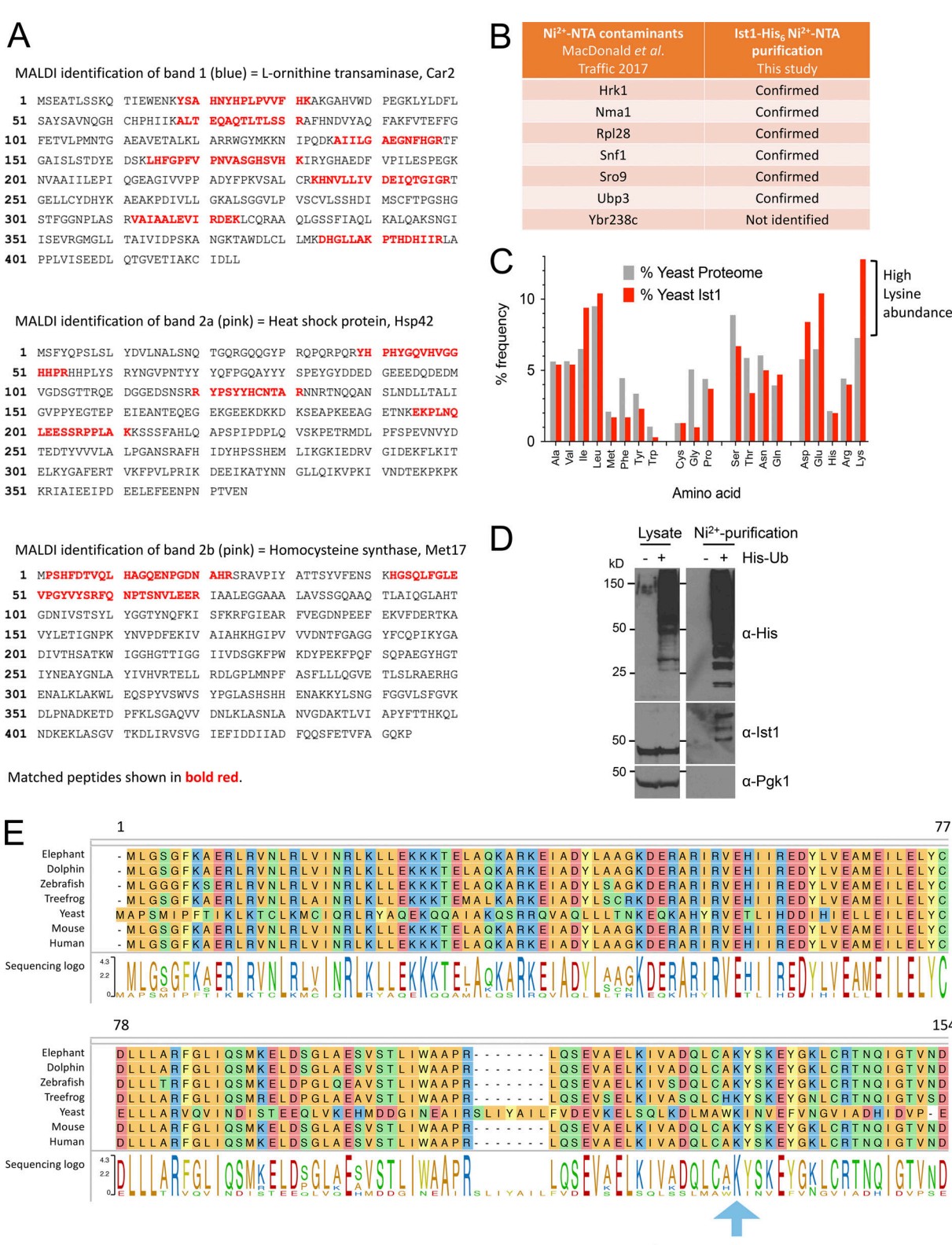

Figure S3. **MS and pull-down controls to show Ist1 is ubiquitinated. (A)** Amino acid sequence of MALDI identified proteins Car2 (top), Hsp4 (middle), and Met17 (bottom) with matched peptide sequences annotated in bold red. **(B)** Table showing previously identified contaminants from yeast lacking His$_6$ tagged proteins purified on a Ni$^{2+}$-NTA column. Table also includes which contaminants were identified by MS from purification of Ist1-HA-His$_6$. **(C)** Histogram depicting average amino acid percentage distribution in yeast proteome (gray) and Ist1 (red). **(D)** WT and His$_6$-ubiquitin expressing cells were grown to log phase before ubiquitinated proteins were isolated from a denatured lysate on Ni$^{2+}$-NTA beads. Original lysates left and purified samples (right) were analyzed by SDS-PAGE followed by immunoblot with the indicated antibodies. **(E)** Ist1 amino acid sequence (from residue 1–154 from *S. cerevisiae* yeast) alignment across species indicated with the highly conserved lysine residue at position 135 (blue arrow).

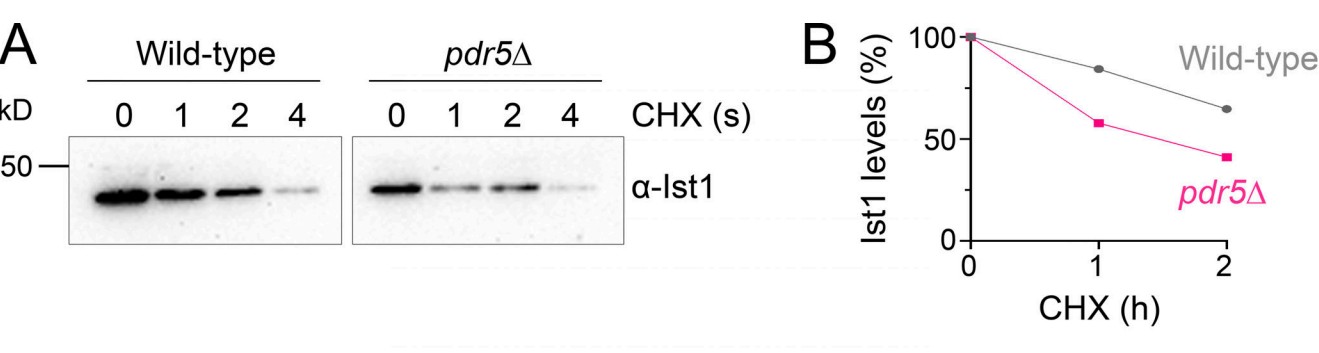

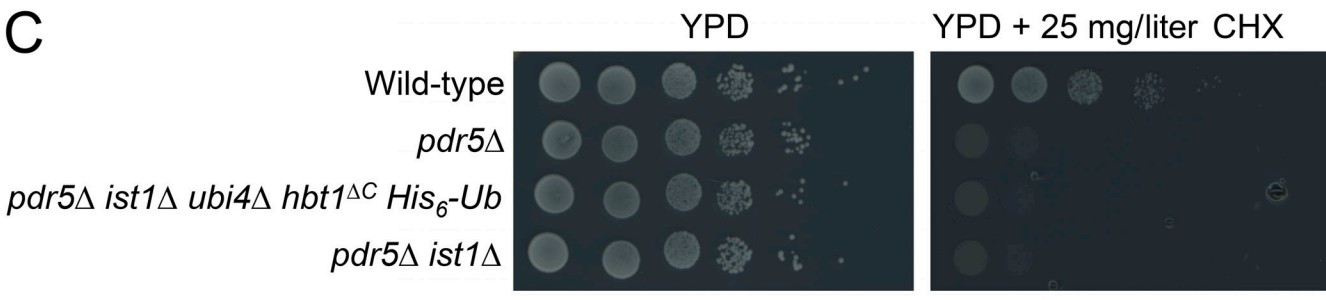

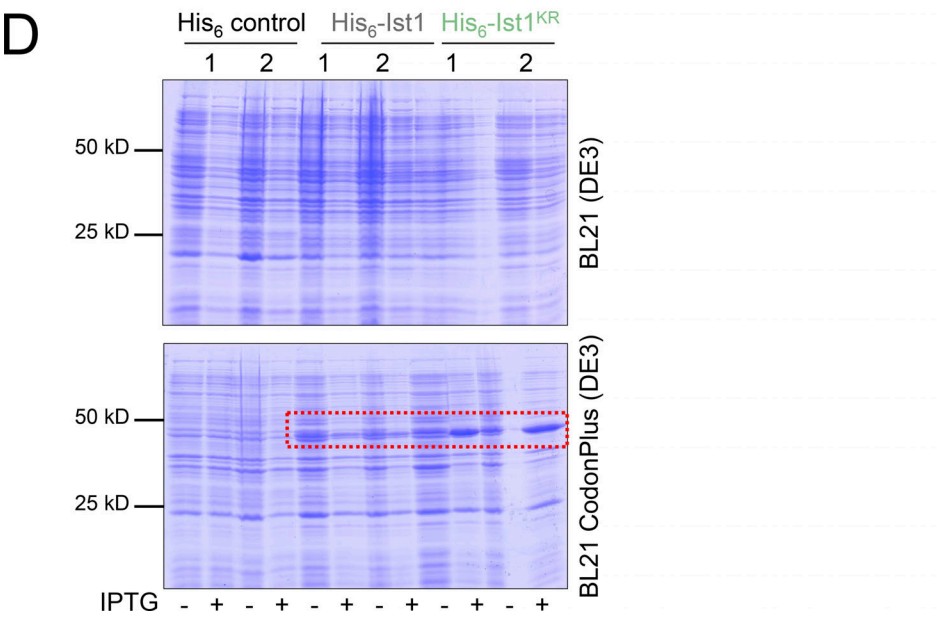

Figure S4. **Assessments of Ist1 stability. (A)** Cycloheximide (CHX) chase experiments using WT and *pdr5Δ* cells grown to mid-log phase before exposure to 25 mg/l cycloheximide for denoted time before harvesting and immunoblot with α-Ist1 antibodies. **(B)** Graph of Ist1 stability following cycloheximide chase at indicated times in WT (gray) and *pdr5Δ* (pink) cells. **(C)** Growth assays in WT, *pdr5Δ*, *pdr5Δ ist1Δ hbt1^{ΔC} His_6-Ub*, and *pdr5Δ ist1Δ* yeast strains grown to exponentially dividing phase then spotted on YPD (left) and YPD containing 25 mg/liter cycloheximide (right). **(D)** SDS-PAGE Coomassie stained gels of lysates from BL21(DE3) (upper) or BL21 CodonPlus(DE3) (lower) *E.coli* strains expressing His_6 control, His_6-Ist1, or His_6-Ist1^{KR} under *T7* promoter control. For each plasmid, two clones are shown from lysates grown to $OD_{600}$ = 0.6, which were grown 16 h at 15°C –/+ 0.5 mM IPTG.

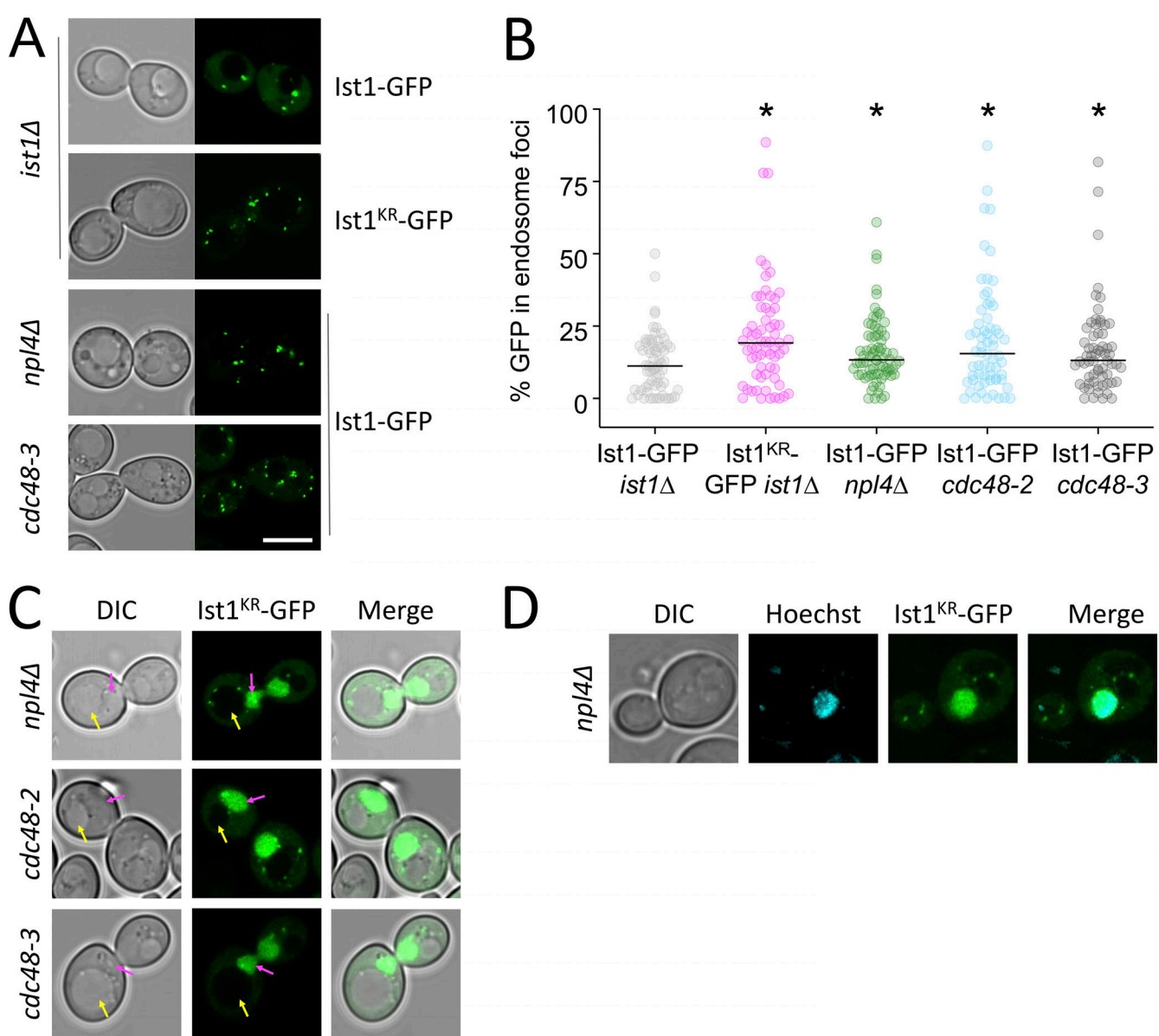

Figure S5. **Ist1 localizations in vivo. (A)** Indicated strains expressing Ist1$^{WT}$-GFP and Ist1$^{KR}$-GFP were imaged by Airyscan microscopy. **(B)** The percentage GFP signal in endosomal foci quantified as a percentage of total cellular fluorescence. **(C)** Ist1$^{KR}$-GFP mislocalizes to the nucleus (pink arrow or Hoechst stain) in indicated mutants. **(D)** Airyscan microscopy used to localize indicated fluorescent proteins in WT cells. Scale bar, 5 μm.

Video 1. **Mup1-GFP + Vps4-mCherry methionine pulse (short imaging intervals).** Frame rate, 5 frames per second (fps).

Video 2. **Mup1-GFP + Sec7-mCherry methionine pulse (short imaging intervals).** Frame rate, 7 fps.

Video 3. **Mup1-GFP + Vps4-mCherry methionine pulse-chase (long imaging intervals).** Frame rate, 7 fps.

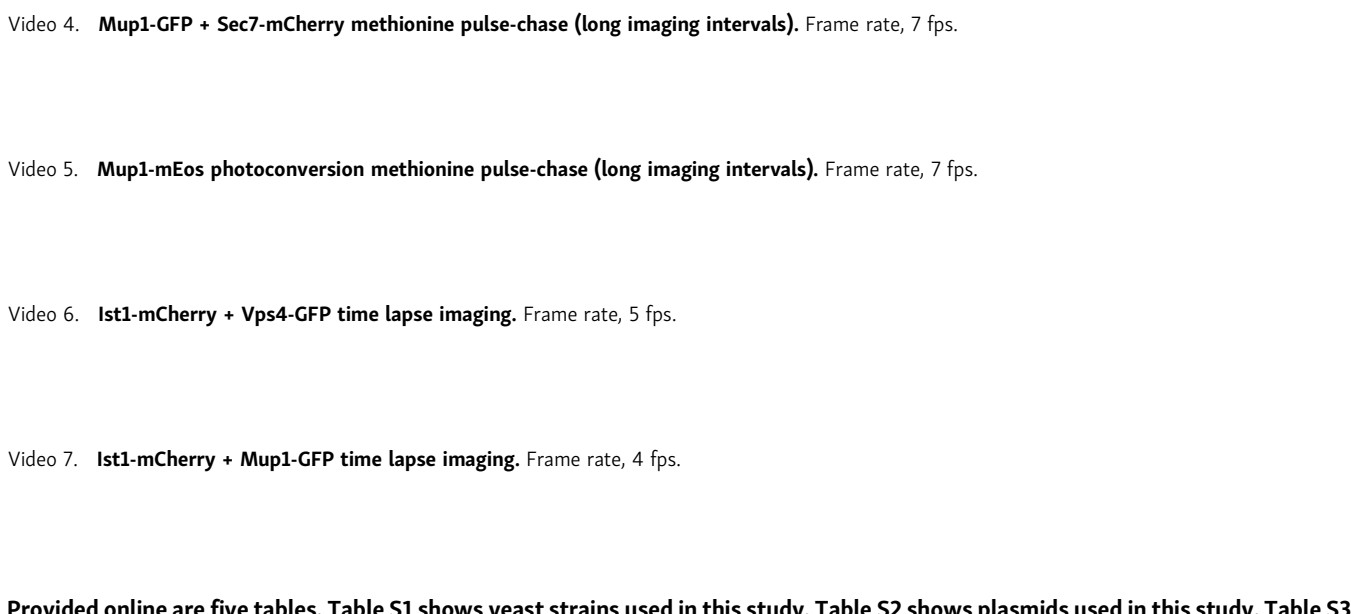

Video 4. **Mup1-GFP + Sec7-mCherry methionine pulse-chase (long imaging intervals).** Frame rate, 7 fps.

Video 5. **Mup1-mEos photoconversion methionine pulse-chase (long imaging intervals).** Frame rate, 7 fps.

Video 6. **Ist1-mCherry + Vps4-GFP time lapse imaging.** Frame rate, 5 fps.

Video 7. **Ist1-mCherry + Mup1-GFP time lapse imaging.** Frame rate, 4 fps.

**Provided online are five tables. Table S1 shows yeast strains used in this study. Table S2 shows plasmids used in this study. Table S3 lists primary antibodies. Table S4 shows statistical analyses. Table S5 shows peptides identified by MS.**

