## [Peer Review File · The Journal of Cell Biology]

Recycling of cell surface membrane proteins from yeast endosomes is regulated by ubiquitinated Ist1

Kamilla Laidlaw, Grant Calder, and Chris MacDonald

Corresponding Author(s): Chris MacDonald, University of York

Review Timeline:

Submission Date:	2021-09-28
Editorial Decision:	2021-10-29
Revision Received:	2022-07-28
Editorial Decision:	2022-08-17
Revision Received:	2022-08-19

Monitoring Editor: Elizabeth Miller

Scientific Editor: Andrea Marat

Transaction Report:

DOI: <https://doi.org/10.1083/jcb.202109137>

October 29, 2021

Re: JCB manuscript #202109137

Dr. Chris MacDonald
University of York
Department of Biology
Office Biology / J001
Wentworth way
York, --- YO10 5DD
United Kingdom

Dear Dr. MacDonald,

Thank you for submitting your manuscript entitled "Recycling of cell surface membrane proteins from yeast endosomes is regulated by ubiquitinated Ist1". Your manuscript has been assessed by expert reviewers, whose comments are appended below. Although the reviewers express potential interest in this work, significant concerns unfortunately preclude publication of the current version of the manuscript in JCB.

You will see that the reviewers appreciate that your study provides potentially important insight into the separation of Golgi and endosomes in yeast, and as such is of high interest to the readership of JCB. However, they have substantial concerns regarding some of the data on Ist1, which we agree is fundamental to address. Therefore, we invite you to submit a revised and expanded manuscript addressing their concerns in full as an Article, as opposed to a short Report.

Essential major revisions include:

- A better description and characterisation of the Ist1-KR mutant (all reviewers)
- Ruling out misfolding effects for this mutant (all reviewers)
- More clearly placing Ist1 in the Cdc48-Npl4 pathway (all reviewers)
- Substantiating the ubiquitination status of Ist1 (reviewer # 3)

In addition, we expect that you will be able to address all of the remaining reviewer comments in your revised manuscript.

Please let us know if you are able to address the major issues outlined above and wish to submit a revised manuscript to JCB. We highly recommend submitting a revision plan, which we may discuss with the reviewers, so that we may provide feedback and help ensure that the revisions will satisfy the reviewer concerns. Note that a substantial amount of additional experimental data likely would be needed to satisfactorily address the concerns of the reviewers. As you may know, the typical timeframe for revisions is three to four months. However, we at JCB realize that the implementation of social distancing measures that limit spread of COVID-19 also pose challenges to scientific researchers. Therefore, JCB has waived the revision time limit. Please note that papers are generally considered through only one revision cycle, so any revised manuscript will likely be either accepted or rejected.

If you choose to revise and resubmit your manuscript, please also attend to the following editorial points. Please direct any editorial questions to the journal office.

GENERAL GUIDELINES:

Text limits: Character count is < 40,000, not including spaces. Count includes title page, abstract, introduction, results, discussion, acknowledgments, and figure legends. Count does not include materials and methods, references, tables, or supplemental legends.

Figures: Your manuscript may have up to 10 main text figures. To avoid delays in production, figures must be prepared according to the policies outlined in our Instructions to Authors, under Data Presentation, <https://jcb.rupress.org/site/misc/ifora.xhtml>. All figures in accepted manuscripts will be screened prior to publication.

Supplemental information: There are strict limits on the allowable amount of supplemental data. Your manuscript may have up to 5 supplemental figures. Up to 10 supplemental videos or flash animations are allowed. A summary of all supplemental material should appear at the end of the Materials and methods section.

Please note that JCB now requires authors to submit Source Data used to generate figures containing gels and Western blots with all revised manuscripts. This Source Data consists of fully uncropped and unprocessed images for each gel/blot displayed in the main and supplemental figures. Since your paper includes cropped gel and/or blot images, please be sure to provide one Source Data file for each figure that contains gels and/or blots along with your revised manuscript files. File names for Source Data figures should be alphanumeric without any spaces or special characters (i.e., SourceDataF#, where F# refers to the associated main figure number or SourceDataFS# for those associated with Supplementary figures). The lanes of the gels/blots should be labeled as they are in the associated figure, the place where cropping was applied should be marked (with a box), and molecular weight/size standards should be labeled wherever possible.

If you choose to resubmit, please include a cover letter addressing the reviewers' comments point by point. Please also highlight all changes in the text of the manuscript.

Regardless of how you choose to proceed, we hope that the comments below will prove constructive as your work progresses. We would be happy to discuss them further once you've had a chance to consider the points raised. You can contact the journal office with any questions, cellbio@rockefeller.edu or call (212) 327-8588.

Thank you for thinking of JCB as an appropriate place to publish your work.

Sincerely,

Elizabeth Miller, PhD
Monitoring Editor

Andrea L. Marat, PhD
Senior Scientific Editor

Journal of Cell Biology

Reviewer #1 (Comments to the Authors (Required)):

Summary

This manuscript explores an exciting area of biology and, if the findings are correct, defines a novel pathway for recycling proteins to the cell surface in the well-tread model system *S. cerevisiae*. The foundation of the paper lies on careful observation of the methionine permease Mup1 and its trafficking post short pulses of methionine in the growth medium, rather than the prolonged methionine exposures typically done to study endocytosis and vacuolar trafficking of Mup1. The authors demonstrate that in response to these short pulses of methionine, we do not see free GFP accumulating in the vacuole, as we would with prolong met incubations and using an elegant photo-conversion approach (mEos tagged Mup1) the authors can show that the puncta of Mup1 returning to the PM is coming from the photoconverted pool of Mup1 that previously resided at the cell surface. The authors spend considerable effort demonstrating that the Mup1 puncta observed post short met incubations co-localize with Vps4/endosome positive compartments and not with Sec7/Golgi compartments. This point is interesting as there has been considerable discussion and literature describing the separation - or lack thereof -- for Golgi and endosome compartments in yeast. Thus the authors conclude that the recycled Mup1 is returning from endosomes and not from the Golgi. In an earlier screen for factors influencing protein recycling, Ist1 was identified. Loss of this factor impairs Mup1 recycling and the data supporting this conclusion are convincing. However, the model for the involvement of Npl4, ubiquitination of Ist1 and Cdc48 are less compelling. In particular, the ist4-KR mutation is not well-defined in the manuscript and the functional implications (ie. loss of ubiquitination vs a non-functional protein) are difficult to tease apart. Added orthogonal approaches, a more tailored suite of Ist1 mutations that are demonstrated to ablate ubiquitination of Ist1 would be more convincing. Further information on the link between Ist1 and Npl4 would strengthen the paper as well.

Major points

1. The photoconversion trick to look at Mup1 at the surface post a brief Met pulse is very elegant. However, given this approach and the 4D imaging being performed, it would be nice to see some single particle tracking so that we could monitor the lifetime of a particular 'patch' of Mup1 as it goes from being internalized to recycled back to the cell surface. You could even provide the amount of time for how long these events take. In figure 2G and movie S1, the Mup1-mEos dot that is 'there and then gone' could be recycling but it would be more compelling to track the puncta's return to the surface. Since you have already done the

4D Airyscan imaging this should be feasible, and it seems a missed opportunity not to provide these analyses. In addition, it would be wonderful to have some idea of the frequency of these recycling events if possible.

2. The mutations comprising Ist1KR are not well defined. The authors indicate that there are 50 lysines and arginines in the protein and that they have performed denaturing Ub purification to show that Ist1 is ubiquitinated (Fig4H), but at what sites? Is Ist1KR mutant lysine-less, as might be inferred by one sentence in the paper? If that is the case, perhaps the protein is non-functional because its structure is disrupted by having >35 lysines mutated? Supplemental Table 2 also has no information on how this mutant was made. To make the distinctions being sought (Npl4 binds Ub'ed Ist1 to recruit Cdc48 and stimulate recycling) more information on the Ist1KR mutation is needed. Is this a functional allele? Perhaps it is misfolded itself and that is why there is so much less of it around in cells (Figure 4K). This possibility should be explored. Are there other functional readouts that could be used to assess the existing Ist1KR mutant's function or is there a more modest mutation that can be made to disrupt ubiquitination while leaving Ist1 function intact? Ub-dependent co-purification of Ist1 and Npl4 would also make the case for the model proposed more robust.

3. The elevated Ist1 levels observed under steady state in *npl4Δ* and Cdc48-hypomorphic mutants is in line with the idea that Npl4 and Cdc48 control Ist1 abundance and stability. However, it would be a more compelling piece of data if a time-course post cycloheximide addition was performed to show that Ist1 is truly more stable in *npl4Δ* cells. To help confirm that Ist1 and Npl4 operate in the same pathway, analyses of an *ist1Δ npl4Δ* double mutant should reveal that they have no additive effects. This seems like a minimal additional analysis to help support your functional model. In addition, if Npl4 is recognizing ubiquitinated Ist1 to disassemble (degrade?) it, then why is the Ist1KR mutation of lower abundance than the WT Ist1 control (Figure 4K)? One might expect the mutant that disrupts Ist1 ubiquitination to be stabilized, but for this mutation that does not appear to be the case. The quantification for this imaging (Figure 4M) is therefore somewhat surprising since we anticipate that the Ist4KR mutant will be of lower abundance (due to the dramatic reduction via immunoblotting) but that is not what is observed. Are all the Ist1KR puncta truly endosomes or could they be aggregated and unfolded protein? An experiment to examine the colocalization of this mutant with Vps4 would help ensure that the puncta examined are indeed endosomes.

Minor points

Why don't the times shown in Figure 3A and 3B correspond across the top and bottom panels? It would be useful to be comparing the co-localization of Mup1 internalized puncta with Sec7 vs Vps4 at the same time points, especially given the dynamic nature of the process being tracked.

Invoking the idea that the co-localization of Mup1 with Sec7 positive compartments is an artifact of deconvolution is somewhat troubling (page 3; end of 2nd paragraph). Why would this colocalization be an artifact while other colocalizations are not? Perhaps this is an even faster recycling pool?

Statistical tests used should be indicated in the figure legend for all cases, especially since they are not listed in the methods. Examples where this is not the case occur in Figure 1E, 2D, and 4M. The Supplement S4 provides the p-values but still doesn't indicate the statistical test performed.

The model in 4A for how Ist1 is functioning - in the tubulation and scission steps for recycling vesicle formation - seems overstated at this stage. There is no data in this paper to support that view of Ist1 function at a biochemical level. It would be good to emphasize that the model for Ist1 is speculative at this stage.

Reviewer #2 (Comments to the Authors (Required)):

The manuscript submitted by Kamilla Laidlaw and co-workers describes that the recycling of cell surface membrane proteins from yeast endosomes is regulated by ubiquitinated Ist1.

The manuscript contains several carefully executed and analyzed imaging experiments and clearly demonstrates that endocytic cargoes such as Snc1 and Snc2 sort to the Golgi form where they recycle back to the PM. Other endocytic cargoes such as nutrient transporters Mup1 or Fur4 are sorted into endosomes that acquire the ESCRT machinery for sorting through multivesicular body sorting into the vacuole.

The paper also describes a function of Ist1 (an ESCRT-III component), the ubiquitination of Ist1 and Cdc48-Npl4 dependent 'depolymerization' of ubiquitinated Ist1 for the recycling of FM4-64 and Ste3-GFP-DUb. While these observations are certainly interesting, several aspects of this part of the paper, (the Ist1 - ubiquitination - Cdc48 axis), are unfortunately incomplete at the moment and several points must be addressed to substantiate the certainly attractive model: ubiquitinated Ist1 polymers recycle cargo and are disassembly by Cdc48-Npl4.

Major points:

Figure 1: I would suggest to tune down the conclusions in the model in Figure 1K. There is no evidence that Mup1-GFP returns from a Vps4 positive endosome back to the PM. In my view, the data shows that in response to methionine treatment Mup1-GFP co-localizes with Vps4 positive MVBs and is transported into the vacuole (most likely via these Vps4 positive MVBs). In contrast, Snc1/2 recycle between the PM and the Golgi.

Figure 2: The pulse chase micro-fluidics experiments are elegant. Yet I find it difficult to conclude that the internalized fraction of Mup1-GFP is recycling back to the PM. Along the same lines, since only a relatively small fraction is internalized, vacuolar GFP clipping might not be detected. Would it be possible to photo-convert the internalized pool and follow it back to the PM? Similarly, the photoconversion experiments in Figure 2 seem to argue against a large fraction of recycling Mup1. If there was indeed lots of recycling, the photoconverted Mup1 should begin to spread all over the PM after methionine pulses (not be lateral diffusion).

Figure 3: Is the degradation / recycling of Mup1 / Fur4 affected in Ist1 mutants?

Figure 4: panel H demonstrates that Ist1 can be co-purified with His-Ub under denaturing conditions. This is a strong indication that Ist1 is ubiquitinated. To move forward, the authors generate an Ist1-KR mutant which they claim is resistant to ubiquitination. (i) It is unclear from the text what the Ist1-KR mutant is? Are all 38 lysine residues mutated to arginine? (ii) I believe it is essential to demonstrate that that Ist1-KR mutant is no longer ubiquitinated. (iii) Does the Ist1-KR mutant still interact with its cognate partner proteins (ESCRT-III subunits and Vps4). (iv) Does Ist1-KR no longer interact with Cdc48-Npl4 (or in other words does Ist1 interact with Cdc48-Npl4?). (v) Although difficult, I think for a publication in JCB it would be important to identify the ubiquitinated lysine residues and make more specific Ist1-KR mutants.

The authors suggest that ubiquitinated Ist1 is released from endosomes in a Cdc48-Npl4 dependent manner. These conclusions are based on imaging data and could eventually be supported by subcellular fractionation experiments. In this model it remains puzzling how Ist1-KR can end up in the nucleus upon Cdc48 inactivation. If the model is correct, I would have expected Ist1-KR to be fully locked on endosomes upon Cdc48 inactivation, yet it targets to the nucleus (for nuclear quality control / PMID: 30429547?).

Minor point:

There is a typo in the last paragraph of the introduction:

...the pathway used by nutrient transporters for (from) methionine (Mup1) and uracil (Fur4).

Reviewer #3 (Comments to the Authors (Required)):

Laidlaw and colleagues explore the routes by which the membrane proteins Fur4, Mup1 and Ste3 are recycled from endosomes back to the plasma membrane. Their data argue that this is a direct route that bypasses the Golgi and requires Ist1 and surprisingly Cdc48 for recycling. This is not the first paper that argues for a direct endosome to plasma membrane route for recycling proteins (see PMID 15215319) but it does present the most rigorous data I've seen in support of this hypothesis. Similarly, Ist1 was previously implicated in endosomal recycling in animal cells (as opposed to a role in MVB biogenesis like most ESCRT-associated proteins) but observing this requirement in yeast allows application of facile molecular approaches to better understand mechanism. The data implicating Cdc48, its Npl4 adaptor and ubiquitination of Ist1 in its recycling function provides layers of new mechanistic insight. The data presented support a model whereby Npl4 recognition of Ist1-Ub recruits Cdc48 for disassembly of Ist1 oligomers. Precisely how the cargo is selected for transport in this route remains to be determined. I thought the data presented in figures 1 - 3 provide a very rigorous support of a recycling route that does not involve the Sec7 compartment. However, the data in Figure 4 underpinning the new mechanistic insight for how Ist1 works was less convincing.

Primary concerns

1. The evidence that Ist1 ubiquitination is important in the recycling pathway comes from complementation tests with Ist1KR-HA, which carries more than 20 K to R mutations. Whether it is the loss of ubiquitin modification or a perturbation in some other aspect of Ist1 function is unclear. The authors should test Ist1-GFP-DUb and a catalytically dead version of this construct for complementation of *ist1Δ* defects in FM4-64 and/or Fur4 recycling.

2. The data shown in Figure 4H is minimal support to claim that Ist1 is ubiquitinated, which is a major conclusion reflected in the title of the manuscript. The authors could apply a mass spec approach by using a protease other than trypsin to cleave Ist1 to identify ubiquitinated lysines. Conversely, the authors could treat samples comparable to those shown in Fig4H, but isolated under non-denaturing conditions, with or without deubiquitinase and determine if any of the bands detected with Ist1 antibody collapse to a mobility closer to the predicted molecular weight of 34.5 kDa. No data is presented here to convince the reader that the bands shown are actually Ist1 - it would be important to pull down ubiquitinated proteins from an *ist1Δ* strain and probe that sample with anti-Ist1 as a control.

3. The authors provide good genetic support for the role of Npl4 and Cdc48 in recycling, but the model could be reinforced by determining if they can detect the Cdc48-Npl4-Ist1-Ub interactions (as shown in Fig4I) using pulldown approaches.

4. The data shown in Figures 4L and M are not very convincing to argue for a role of Ist1 ubiquitination and Cdc48-Npl4 in regulating endosome dynamics of Ist1. The data may be statistically significant but this does not equate to biological significance. In addition, the method of quantitation was not described for these imaging data. I suggest either removing these results or try to provide another method of evaluation, such as subcellular fractionation by differential centrifugation followed by western blot for Ist1.

5. Figure 4N lacks a catalytically dead control or at least a non-DUb fusion control for comparison.

6. Figure 4L shows punctate localization of Ist1KR-GFP while supplemental figure 4d shows that Ist1-GFP-DUb is nearly all cytosolic when expressed in *ist1Δ* cells but punctate when expressed in WT cells. I don't believe the latter result was described or discussed anywhere in the text. Does this imply a role of Ist1 ubiquitination in assembly or recruitment to the endosome? Does a catalytically dead version of this construct localize to punctae in *ist1Δ* cells?

Minor comments

a. Figure 1F vs 1G. The text states that the DUB fusion is causing an increased localization of Snc1 and Snc2 to the Golgi but the MOC vacuoles look about the same. A MOC of 75% seems inflated for GFP-Snc2 relative to the image shown.

b. I believe coloc 2 in Image J calculates a Manders' colocalization coefficient and not an overlap coefficient. Was a different plugin used? If an MCC, how were negative values obtained in Fig 3C? The Methods section indicates a Pearson coefficient was reported except for timelapse. More clarity in the approach taken is needed.

c. Figure 1 I and J - what was the time of Met or Ura addition?

d. The photoconversion experiments are very cool. Do you see the Mup1-Eos red signal in endosomes come back and merge into the green segment of the plasma membrane? It would be nice to see the fate of those molecules.

e. The first Intro paragraph covers a lot of material could be split into 2 or 3 paragraphs.

f. Owing should be owing on the top of page 4.

RESPONSE TO REVIEWERS

We would like to sincerely thank the reviewers for all the comments we received, it highlighted several key weaknesses of the original paper that we fully agree needed to be addressed. We also appreciate the focussed nature of the comments, this made our job of addressing critical aspects much easier. We have tried to take all comments on board, reconcile any differences of opinion, and consolidate old and new data into a coherent story regarding the organisation of yeast endosomes and the regulation of endosomal recycling. The suggested experiments have essentially all supported the original model. We have also tapered our conclusions to allow alternative views to be considered.

Although the overall model for endosomes and Ist1-recycling remain essentially unchanged, it was suggested that the text and figures be expanded in the format of an article. As such, the conclusions and requested experiments have been laid out in 9 main and 5 supplemental figures. For simplicity, all references herein are to the new figure format. In addition to the specific responses detailed below (some points have been broken down to address individual aspects clearly). A summary of the improvements include:

- We provide detailed colocalization analysis as main **Figure 3D - 3F** from live imaging in real-time (in addition to the steady state analysis previously included).
- We now provide evidence in **Figure 4G** that 1) *ist1*Δ cells are defective in recycling Mup1-GFP and 2) that Mup1-GFP molecules that fail to recycle accumulate in Vps4 endosomes (and not Sec7 Golgi compartments). Both observations connect better to the work proposed earlier in the paper.
- As suggested, demonstrating that combining mutations *ist1* and either *cdc48* or *npl4* show no additive defects in recycling (**Figure 5H - 5I**) better supports the notion that the novel roles of Ist1 and Cdc48/Npl4 in recycling are functionally connected.
- To show Ist1 ubiquitination we created an endogenously tagged Ist1 strain and extensively optimised 1) purification and 2) processing for mass spectrometry to favour identification of Ub~Ist1 peptides (**Figure 6A - 6E, S3A - S3E**). This revealed one definable ubiquitinated peptide (**Figure 6F**), but we also go on to show that purification of ubiquitinated conjugates yields ubiquitinated species of Ist1 that are lost when *IST1* is deleted (**Figure 6G - 6H**).
- We go on to show that the Ist1^{KR} mutant is also defective in recycling Mup1 via Vps4 endosomes (**Figure 7F**).
- The conclusions based on the lysineless (Ist1^{KR}) mutant - which was insufficiently described and does include all 38 lysine residues mutated to arginine - were indeed preliminary. We now show this mutant to be stable *in vivo*: with similar turn-over kinetics with Ist1^{WT} following a cycloheximide chase (**Figure 7B - 7C, S4A - C**) and correct folding suggested by localization to Vps4 endosomes (**Figure 7D - 7E**).
- Furthermore, expression and purification of recombinant Ist1^{WT} and Ist1^{KR} from bacteria showed proteins with similar yield and stability in different buffers, during freeze/thaw cycles, and when quantified by nanoscale differential scanning fluorimetry (**Figure 7H - 7M, S7D**).
- As the levels of Ist1^{KR} are lower than Ist1^{WT}, we employed an inducible expression system to control *IST1* expression to mimic either null, WT and KR levels. This showed that lower levels of Ist1 found in the mutant are sufficient to mediate protein and lipid recycling from endosomes (**Figure 8A - 8E**).
- As KR mutants are inherently problematic (e.g. due to unconventional lysines getting ubiquitinated, large point number of point mutations required, etc) the Piper Lab designed an alternative strategy to inhibit substrate ubiquitination (Stringer & Piper **JCB** 2011). When we applied this DUB-fusion strategy to Ist1 the results are entirely consistent with our model that Ub~Ist1 is required for recycling, including the neat control of expressing a catalytically dead DUB fusion, which functions the same wild-type (**Figure 9A - 9G**).
- This DUB fusion work also suggests that the role of Ist1 in recycling may occur from endosomes that are definably different than MVBs, as we observe non-Golgi Ist1-endosomes that lack Vps4 (**Figures 4A, 9D - 9F, and Movies S6 - S7**).

Reviewer #1 (Comments to the Authors (Required)):

Summary

This manuscript explores an exciting area of biology and, if the findings are correct, defines a novel pathway for recycling proteins to the cell surface in the well-tread model system *S. cerevisiae*. The foundation of the paper lies on careful observation of the methionine permease Mup1 and its trafficking post short pulses of methionine in the growth medium, rather than the prolonged methionine exposures typically done to study endocytosis and vacuolar trafficking of Mup1. The authors demonstrate that in response to these short pulses of methionine, we do not see free GFP accumulating in the vacuole, as we would with prolonged met incubations and using an elegant photo-conversion approach (mEos tagged Mup1) the authors can show that the puncta of Mup1 returning to the PM is coming from the photoconverted pool of Mup1 that previously resided at the cell surface. The authors spend considerable effort demonstrating that the Mup1 puncta observed post short met incubations co-localize with Vps4/endosome positive compartments and not with Sec7/Golgi compartments. This point is interesting as there has been considerable discussion and literature describing the separation - or lack thereof -- for Golgi and endosome compartments in yeast. Thus the authors conclude that the recycled Mup1 is returning from endosomes and not from the Golgi. In an earlier screen for factors influencing protein recycling, Ist1 was identified. Loss of this factor impairs Mup1 recycling and the data supporting this conclusion are convincing. However, the model for the involvement of Npl4, ubiquitination of Ist1 and Cdc48 are less compelling. In particular, the ist4-KR mutation is not well-defined in the manuscript and the functional implications (ie. loss of ubiquitination vs a non-functional protein) are difficult to tease apart. Added orthogonal approaches, a more tailored suite of Ist1 mutations that are demonstrated to ablate ubiquitination of Ist1 would be more convincing. Further information on the link between Ist1 and Npl4 would strengthen the paper as well.

Major points

1. The photoconversion trick to look at Mup1 at the surface post a brief Met pulse is very elegant. However, given this approach and the 4D imaging being performed, it would be nice to see some single particle tracking so that we could monitor the lifetime of a particular 'patch' of Mup1 as it goes from being internalized to recycled back to the cell surface. You could even provide the amount of time for how long these events take. In figure 2G and movie S1, the Mup1-mEos dot that is 'there and then gone' could be recycling but it would be more compelling to track the puncta's return to the surface. Since you have already done the 4D Airyscan imaging this should be feasible, and it seems a missed opportunity not to provide these analyses. In addition, it would be wonderful to have some idea of the frequency of these recycling events if possible.

Our lengthy optimization of 4D Airyscan imaging coupled to microfluidic induced transporter recycling has yielded some key discoveries for this paper. However, even with this technology, it is currently very difficult to track single particles across a 3D volume during time lapse imaging.

We find Airyscan imaging, or even a new 4x faster imaging technique using Airyscan2, imaging modes were better than structured illumination (even in apotome and leap modes provided by the Zeiss Elyra7) due to the low levels of signal from these proteins expressed at endogenous levels. However, even with Airyscan there are significant constraints, chiefly: 1) the small size of these fast-moving molecules and the time it takes to scan (especially for 3D volume) and 2) photobleaching, which is much worse for mEos, after the fairly aggressive photoconversion procedure that requires a near ultraviolet laser.

By focussing scan speed to only a 2D volume and the 4-fold increase in rate using Airyscan2, we can increase speed of imaging to track particles. This approach is inherently difficult, as it relies on particles remaining in this limited imaging volume. However, we did capture Mup1-GFP recycling from Ist1-marked endosomes back to the surface and included these observations (**Movie S7**).

2A - The mutations comprising Ist1KR are not well defined.

There are several points here that require clarification. We are thankful for all reviewers highlighting these aspects of the work and I hope the new data, explanations and revised text resolve these issues (see below).

2B - The authors indicate that there are 50 lysines and arginines in the protein and that they have performed denaturing Ub purification to show that Ist1 is ubiquitinated (Fig4H), but at what sites?

There are 38 lysine residues in yeast Ist1, representing ~13% of the amino acid sequence, that could serve as substrate residues for ubiquitination. Decades of global proteomics in yeast, including efforts dedicated to enriching and identifying ubiquitinated peptides, have failed to reveal a single ubiquitinated lysine in Ist1. We believe this is due to the high % of Arg and Lys residues in Ist1, which are sites for protease digestion by the most

commonly used enzymes (trypsin, and to a lesser degree LysC) used to generate peptides for mass spectrometry. This issue is exacerbated by ubiquitinated species typically representing only a tiny fraction of a cellular protein pool and lysis procedures being prone to substrate deubiquitination (MacDoanld *et al.*, **Traffic** 2017).

To identify ubiquitinated Ist1 peptides we generated an endogenously tagged version of Ist1 and performed a two-step denatured purification from several litres of culture (**Figure 6C**). However, this enriched sample did not give obviously detectable protein bands on a gel when cut out and analysed by mass-spec (**Figure 6D**). We therefore analysed the entire pool of purified proteins. In a bid to generate more “mass spec-able” Ist1 peptides we used two sequential proteases for peptide generation (as suggested by reviewer 3) that were selected to maximise the coverage of hypothetical ubiquitinated peptides. We also used a propionylation modification step (Drury *et al* **Biochemical J.** 2012) to enrich charged peptides that contain lysine(s) during processing. Analysis was performed on a high resolution Orbitrap Fusion Tribrid mass spectrometer. Finally, we optimised chromatography and ran multiple long gradients, then used search algorithms optimised for identifying ubiquitinated peptides (PEAKS StudioX Pro and Byonic Protein Metrics software). This effort identified the highly conserved Ist1-K135 as ubiquitinated (**Figure 6C, S3E**).

However, as mentioned in my revision plan, one massive concern with bottom-up proteomics is that we can be confident in what we do see but not what we don't. It is highly possible that other Ist1 lysine residues are ubiquitinated, and we believe even if methods did exist to unequivocally identify all Ist1 residues ubiquitinated *in vivo* then targeted KR point mutants would only result in the ubiquitination of alternative, non-canonical lysines (especially given so many possibilities across the protein).

I contributed to the discovery that lysine prioritisation occurs when E3-ligases ubiquitinate substrates *in vitro* (Kamadurai, **eLife** 2013). Since then, our unpublished experiments demonstrate that although lysines are also prioritised *in vivo*, mutants targeted at blocking ubiquitination should be interpreted with caution. Even random, sub-optimally positioned lysines can be ubiquitinated to very significant levels (**Rebuttal Figure R1**). Consider this in the context of Ist1, which has 38 lysines peppered along its relatively few 298 amino acids. Furthermore, we see no evidence that a single point mutation of lysine residues tested is sufficient to impact recycling (**Figure S2A**).

Rebuttal Figure 1: The wild-type (WT) cargo Sna3 is ubiquitinated at its only two endogenous cytosolic lysines (K19 and K125). Mutation of both these lysines to arginine (K19R K125R) entirely blocks ubiquitination. In the K19R K125R background, reintroducing lysine at the endogenous K125 position allows the most robust ubiquitination (i.e. the lysine position is prioritised) but critically, randomly introducing lysines within Sna3 results in significant (between ~40 and ~70%) levels of ubiquitination!

Given this, we think the relevant question is probably not “what sites are ubiquitinated?” but “does Ist1 have many redundant ubiquitination sites because ubiquitination is critical for its function?” Indeed, the lysine content of Ist1 is almost double the average across the yeast proteome (**Figure S3C**). We know Ist1 is ubiquitinated - using distinct methods (**Figure 6A - 6H**), and that Ist1 is degraded via the ubiquitin proteasome systems (Jones *et al.*, **Traffic** 2012). Beyond this, we also use different approaches and controls to demonstrate this ubiquitination is critical for the role of Ist1 in recycling (**Figures 7A - 7M, 8A - 8E, and 9A - 9G**).

2C - Is Ist1KR mutant lysine-less, as might be inferred by one sentence in the paper?

We apologise for the insufficient description of the Ist1^{KR} mutant - it is 38 lysines mutated to arginine. We have now described this mutant in the results (page 5, line 389) and supplemental methods section (**Supplemental Table T2**) in proper detail, and we have characterised this mutant in various additional ways, discussed below.

2D - If that is the case, perhaps the protein is non-functional because its structure is disrupted by having >35 lysines mutated? Supplemental Table 2 also has no information on how this mutant was made. To make the

distinctions being sought (Npl4 binds UB'ed Ist1 to recruit Cdc48 and stimulate recycling) more information on the Ist1^{KR} mutation is needed. Is this a functional allele? Perhaps it is misfolded itself and that is why there is so much less of it around in cells (Figure 4K). This possibility should be explored.

The Ist1^{KR} mutant appears to fold correctly, as assessed by nanoscale differential scanning fluorimetry (nanoDSF) analysis of recombinantly purified protein (Figure 7H - 7M). Although the nanoDSF data was extremely convincing we sought alternative approaches to confirm this, which were unfortunately unsuccessful.

The purified protein (both WT and KR behaved the same) was not amenable to concentration to levels sufficient for SEC-MALLS, despite our optimisation to give significant increases in yield (Figure S4D). Similarly, recombinantly purified Ist1 (both WT and KR) is sensitive to salt conditions (Dave Katzmann personal communication and Tan et al., JBC 2015) and was not amenable to dialysis against buffers that are compatible with circular dichroism. Finally, we labelled WT and KR versions of Ist1 containing a His₆-tag with His-Lite OG488 and performed Flow Induced Dispersion Analysis (FIDA), but we could not generate a parabolic hydrodynamic flow profile above the noise of buffer alone.

We further tested the Ist1^{KR} mutant *in vivo* to also suggests this mutant is folded correctly. Not only does the Ist1^{KR} mutant exhibit similar stability to wild-type following cycloheximide chase (Figures 7B - 7C) but it localizes to Vps4-positive endosomes (Figure 7D) and fails to support recycling from them (Figure 7E). Finally, the lower levels of this Ist1^{KR} mutant were not considered a big concern as they support recycling (Figure 8A - 8E). Collectively, we feel lack of ubiquitinatable lysines is the most likely explanation for why Ist1^{KR} to support recycling (Figure 7F), substantiated by the complementary yet distinct ubiquitin-reversal strategy (Figure 9A - 9G).

2E - Are there other functional readouts that could be used to assess the existing Ist1^{KR} mutant's function or is there a more modest mutation that can be made to disrupt ubiquitination while leaving Ist1 function intact? Ub-dependent co-purification of Ist1 and Npl4 would also make the case for the model proposed more robust.

We don't think more modest mutations of Ist1 are likely to reveal anything interpretable, as discussed above in response to point 2B. However, we have also employed an alternative strategy to reverse ubiquitination (Stringer & Piper JCB 2011) of Ist1 by fusion of a deubiquitinating enzyme, which inhibits endosomal recycling (Figures 9F). The great thing about this result is a very modest mutation can be made, in the catalytic domain of the DUB to kill its activity without modifying Ist1 itself. The catalytically dead fusion to Ist1 has no impact on recycling (Figures 9G).

3A - The elevated Ist1 levels observed under steady state in *npl4Δ* and Cdc48-hypomorphic mutants is in line with the idea that Npl4 and Cdc48 control Ist1 abundance and stability. However, it would be a more compelling piece of data if a time-course post cycloheximide addition was performed to show that Ist1 is truly more stable in *npl4Δ* cells. To help confirm that Ist1 and Npl4 operate in the same pathway, analyses of an *ist1Δ npl4Δ* double mutant should reveal that they have no additive effects. This seems like a minimal additional analysis to help support your functional model.

We considered the absolute levels the best read out for this given the localization (and mislocalization) phenotypes (Figure S5C - S5D) and the relatively little impact cycloheximide had between wild-type and KR versions of Ist1 (Figure 7C - 7D). Thank you for the clever experimental suggestion of creating combination mutants. We found the defects of either *npl4Δ* or temperature sensitive *cdc48* mutants were not exacerbated by further deletion of *IST1* (Figure 5H - 5I) and really think this helps support the model.

3B - In addition, if Npl4 is recognizing ubiquitinated Ist1 to disassemble (degrade?) it, then why is the Ist1^{KR} mutation of lower abundance than the WT Ist1 control (Figure 4K)? One might expect the mutant that disrupts Ist1 ubiquitination to be stabilized, but for this mutation that does not appear to be the case. The quantification for this imaging (Figure 4M) is therefore somewhat surprising since we anticipate that the Ist1^{KR} mutant will be of lower abundance (due to the dramatic reduction via immunoblotting) but that is not what is observed. Are all the Ist1^{KR} puncta truly endosomes or could they be aggregated and unfolded protein? An experiment to examine the colocalization of this mutant with Vps4 would help ensure that the puncta examined are indeed endosomes.

We also were surprised that the Ist1^{KR} mutant had lower cellular levels. Of note, Ist1 remains relatively stable even hours after cycloheximide chase (Figure 7C - 7D). Therefore, we speculate that although Ist1 is turned over in a ubiquitin dependent manner, the cycling of ubiquitination (which might be required to depolymerise but not degrade) is tied to its function and not exclusively to regulate protein degradation.

The data in Figure S5A - S5B (previously 4M) represent the differences in cytosolic and endosomally localized puncta as a percentage, not as an absolute quantification across different imaging experiments. Reviewer 3, point 4 also highlighted that these differences are subtle and in response to that we have moved

these observations to the supplemental material for documentation but do not use them for heavy conclusions. Furthermore, the levels of Ist1 are addressed separately now (8A - 8E).

We performed the suggested experiments and find that Ist1^{KR}-GFP puncta colocalizes with Vps4 when quantified (Figure 7D - 7E), pretty much in the same way as Ist1^{WT} (Figure 4A). This suggests the puncta do not represent aggregated unfolded species, which is further supported by the *in vitro* observation that Ist1^{KR} is stable at physiological temperatures similar to Ist1^{WT} (Figure 7H - 7M).

Minor points

4 - Why don't the times shown in Figure 3A and 3B correspond across the top and bottom panels? It would be useful to be comparing the co-localization of Mup1 internalized puncta with Sec7 vs Vps4 at the same time points, especially given the dynamic nature of the process being tracked.

Only a small selection of representative micrographs is included in Figure 3A - 3B. The best comparisons of Mup1 localization with Sec7 and Vps4 is time lapse datasets that include every frame (Movies S2 - S5) and from quantification of colocalization measured at steady state and in real time (Figure 3C - 3F).

5 - Invoking the idea that the co-localization of Mup1 with Sec7 positive compartments is an artifact of deconvolution is somewhat troubling (page 3; end of 2nd paragraph). Why would this colocalization be an artifact while other colocalizations are not? Perhaps this is an even faster recycling pool?

Thank you for pointing this issue out, we do not want readers troubled by this point. I fear my use of the word 'artifact' has somehow suggested that this data cannot be trusted. The wealth of data provided in Figures 1 - 3 and Movies S1 - S5 unequivocally support the model that nutrient transporters recycle independently of transit via the Golgi. The differences in colocalization using different methods very strongly indicate that Mup1 almost exclusively transits Vps4 endosomes and not Sec7 labelled compartments.

The point I was trying to make is that our data do not show "zero" co-localization between Mup1-GFP and Sec7-mCherry. However, on inspection the only time the software identified a small colocalization signal between Mup1 and Sec7 was noticeably when Sec7 was in the proximity of the PM (we - and many labs - have localized Sec7 to numerous puncta dispersed throughout the cell, including some that can be close to the surface). The idea I was trying to invoke that might explain this very specific colocalization is depicted in Rebuttal Figure 2.

Rebuttal Figure 2: One explanation for the small amount of Sec7 colocalization with Mup1 only ever being registered at the surface (so not a recycling population of Mup1) is depicted in this simplified figure. Theoretically, following deconvolution of z-stack images the increased optical thickness of the analysis plane might allow surface localized Mup1 to be incorrectly assumed to colocalize with peripheral Golgi compartments, but only if close to the PM

We present these observations now (Figure 3E and 3F) and include a small discussion of potential reasons for the reader to fully interpret including the possibility that noise (from surface localized Mup1 and not recycling Mup1) might contribute to colocalization values (page 4, line 225).

6 - Statistical tests used should be indicated in the figure legend for all cases, especially since they are not listed in the methods. Examples where this is not the case occur in Figure 1E, 2D, and 4M. The Supplement S4 provides the p-values but still doesn't indicate the statistical test performed.

More information about statistical tests has been added to both the figure legends, the methods and Supplemental Table S4.

7 - The model in 4A for how Ist1 is functioning - in the tubulation and scission steps for recycling vesicle formation - seems overstated at this stage. There is no data in this paper to support that view of Ist1 function at a biochemical level. It would be good to emphasize that the model for Ist1 is speculative at this stage.

This is a valid point. For clarification, the model was not included at this stage to mislead the reader about our yeast observations but rather to summarise what is known about Ist1 function in mammalian cells before

presenting all the later data we provide from yeast. This is now explicitly stated (*page 4, line 250*) when referring to this schematic. We also reinforce the point that the function of mammalian Ist1 in endosomal recycling is exactly what our genetic screen sought to identify in yeast - so the revelation that Ist1 was implicated in endosomal recycling in yeast from a blind survey of 5200 mutants is very encouraging 😊

Reviewer #2 (Comments to the Authors (Required)):

The manuscript submitted by Kamilla Laidlaw and co-workers describes that the recycling of cell surface membrane proteins from yeast endosomes is regulated by ubiquitinated Ist1.

The manuscript contains several carefully executed and analyzed imaging experiments and clearly demonstrates that endocytic cargoes such as Snc1 and Snc2 sort to the Golgi form where they recycle back to the PM. Other endocytic cargoes such as nutrient transporters Mup1 or Fur4 are sorted into endosomes that acquire the ESCRT machinery for sorting through multivesicular body sorting into the vacuole.

The paper also describes a function of Ist1 (an ESCRT-III component), the ubiquitination of Ist1 and Cdc48-Npl4 dependent 'depolymerization' of ubiquitinated Ist1 for the recycling of FM4-64 and Ste3-GFP-DUb. While these observations are certainly interesting, several aspects of this part of the paper, (the Ist1 - ubiquitination - Cdc48 axis), are unfortunately incomplete at the moment and several points must be addressed to substantiate the certainly attractive model: ubiquitinated Ist1 polymers recycle cargo and are disassembly by Cdc48-Npl4.

Major points:

1 - Figure 1: I would suggest to tune down the conclusions in the model in Figure 1K. There is no evidence that Mup1-GFP returns from a Vps4 positive endosome back to the PM. In my view, the data shows that in response to methionine treatment Mup1-GFP co-localizes with Vps4 positive MVBs and is transported into the vacuole (most likely via these Vps4 positive MVBs). In contrast, Snc1/2 recycle between the PM and the Golgi.

We thought it important to include a summary model at this stage that documents all the results in **Figure 1**, but also set up the hypotheses that are later tested and (we hope you agree) largely confirmed in the later parts of the manuscript. The data in Figure 1 support the idea that Snc1/Snc2 recycles via the Golgi in a cargo ubiquitination dependent manner (Xu *et al.*, *eLife* 2017, Best *et al.*, *MBoC* 2020) but that Mup1/Fur4 recycle without transiting Golgi compartments in a cargo *deubiquitination* manner (MacDonald *et al.*, *EMBO Reports* 2012, MacDonald *et al.*, *Developmental Cell* 2015, MacDonald & Piper *JCB* 2017).

Given this, in our revised manuscript we have therefore not modified the model (**Figure 1K**), which is hypothetical and is referred to in the text by "we propose". Maybe further guidance from the reviewers / editors about whether this rationale is valid would be useful. If this schematic is considered a major issue, we could maybe remove the arrows back to the surface and discuss only internalisation of each cargo when proposing recycling routes? I am a little worried that although I could tone this down and easily modify the figure, this might do a disservice to the data. I think it is critical to highlight *the opposite nature* of recycling between Snc1/Snc2 and Mup1/Fur4, with regards to ubiquitination/deubiquitination (**Figures 1A and 1B**), mother/daughter cell polarisation (**Figures 1C - 1E**) and Sec7/Vps4 localization (**Figures 1F - 1J**).

It is also important to highlight here that we do not believe: 1) internalised Mup1 traffics to the vacuole if the ubiquitin-signal for degradation is removed (either using our methionine pulse-chase protocol or DUB-fusion method) or that all Vps4 positive compartments are MVBs (Adell *et al.*, *eLife* 2017).

2 - Figure 2: The pulse chase micro-fluidics experiments are elegant. Yet I find it difficult to conclude that the internalized fraction of Mup1-GFP is recycling back to the PM. Along the same lines, since only a relatively small fraction is internalized, vacuolar GFP clipping might not be detected. Would it be possible to photo-convert the internalized pool and follow it back to the PM? Similarly, the photoconversion experiments in Figure 2 seem to argue against a large fraction of recycling Mup1. If there was indeed lots of recycling, the photoconverted Mup1 should begin to spread all over the PM after methionine pulses (not be lateral diffusion).

We did try to photo-convert an internal pool of Mup1 however there are significant technical issues that severely limit our ability to achieve this. Some of these issues are discussed in response to Reviewer 1, Point #1. Beyond this, the ability to photoconvert a fast moving (in x-, y- and z- plane) intracellular spot of Mup1-GFP is essentially impossible, as we do not know in advance when one will be observed in a plane of focus for us to target (and it would have moved before it could be converted). There is not sufficient temporal or spatial resolution to track particles and then also photoconvert and continue tracking. Indeed, you will notice the time frames are

already greatly limited following capture of 1) pre-conditions, 2) pulse-chase induced recycling, 3) photoconversion in real time from single experiments (Figures 2G).

3 - Figure 3: Is the degradation / recycling of Mup1 / Fur4 affected in Ist1 mutants?

We did not assess the degradation of nutrient transporters in *ist1Δ* cells, which has been documented previously (Dimaano *et al.*, MBoC 2008; Rue *et al.*, MBoC 2008). We did pursue the idea of assessing recycling and found recycling of Mup1 is impaired in *ist1Δ* cells, with the transporter accumulating in Vps4-endosomes, and distinct from Sec7-positive compartments (Figure 4G). This was a great suggestion that connects more directly to the localization studies in Figures 1 - 3. We also expanded this idea and show that Ist1^{WT}, but not the lysisneless mutant, can rescue Mup1-GFP recycling from Vps4 endosomes (Figure 7F).

4 - Figure 4 - panel H demonstrates that Ist1 can be co-purified with His-Ub under denaturing conditions. This is a strong indication that Ist1 is ubiquitinated. To move forward, the authors generate an Ist1-KR mutant which they claim is resistant to ubiquitination.

These are all valid concerns, solidified by them being raised by other reviewers. We have addressed each point with more data and explained in detail in the response to Reviewer 1, points 2A - 2E above. In case anything from the rebuttal statement above has not been covered, I will briefly summarise.

(i) It is unclear from the text what the Ist1-KR mutant is? Are all 38 lysine residues mutated to arginine?

Yes, now detailed in the text and Supplemental Table T2.

(ii) I believe it is essential to demonstrate that that Ist1-KR mutant is no longer ubiquitinated.

We now show that lysisneless Ist1^{KR} (Figures 7F - 7G) and enforced deubiquitination of Ist1 (Figures 9F - 9G) is defective for recycling. These are both established and entirely distinct methods to create proteins that are no longer ubiquitinated and are consistent with all other observations in the paper and beyond. Furthermore, the difficulties assessing ubiquitination of Ist1 are discussed at length in the manuscript, the new data, and our response to Reviewer 1, point 2B above.

(iii) Does the Ist1-KR mutant still interact with its cognate partner proteins (ESCRT-III subunits and Vps4).

We find Ist1^{KR} behaves like Ist1^{WT} in many ways, including colocalization with Vps4 (Figure 7D - 7E).

(iv) Does Ist1-KR no longer interact with Cdc48-Npl4 (or in other words does Ist1 interact with Cdc48-Npl4?).

We were unable to biochemically test this due to technical difficulties in immunoprecipitations, so we are very careful to explain that the model is speculative before providing evidence in support of the speculative model and discussing limitations.

(v) Although difficult, I think for a publication in JCB it would important to identify the ubiquitinated lysine residues and make more specific Ist1-KR mutants.

As described above in response to reviewer 1, point 2B, despite great effort, expense and some discoveries towards this goal, a full survey of ubiquitinated Ist1 is simply not feasible with current technology. We further comment that the extent of potential lysine substrate residues in Ist1 for ubiquitination: 1) makes covering this protein by mass spec incredibly difficult, 2) that a minimal KR-mutant that lacks sufficient ubiquitination to test function might never be feasibly identified and/or testable *in vivo*, 3) and that maybe Ist1 has so many potential ubiquitinatable lysine residues to provide redundant modes of ubiquitination. As we show through different approaches that ubiquitination is critical for Ist1 regulation of recycling, this latter speculation is conceptually attractive.

5 - The authors suggest that ubiquitinated Ist1 is release from endosomes in a Cdc48-Npl4 dependent manner. These conclusions are based on imaging data and could eventually be supported by subcellular fractionation experiments. In this model it remains puzzling how Ist1-KR can end up in the nucleus upon Cdc48 inactivation. If the model is correct, I would have expected Ist1-KR to be fully locked on endosomes upon Cdc48 inactivation, yet it targets to the nucleus (for nuclear quality control / PMID: 30429547?).

We agree that there are multiple interpretations of the mislocalization phenotype, which was unexpected and is not simple to rationalise. In line with reviewer 3, point 4 we have toned down the interpretations of these observations and concentrate on the fact that Ub~Ist1 and Npl4/Cdc48 are required for recycling. There is some evidence that they are functionally related (**Figures 5H - 5I, S5A - S5B**) but that other interpretations are available.

Minor point:

There is a typo in the last paragraph of the introduction:
....the pathway used by nutrient transporters for (from) methionine (Mup1) and uracil (Fur4).
Got this, thank you!

Reviewer #3 (Comments to the Authors (Required)):

Laidlaw and colleagues explore the routes by which the membrane proteins Fur4, Mup1 and Ste3 are recycled from endosomes back to the plasma membrane. Their data argue that this is a direct route that bypasses the Golgi and requires Ist1 and surprisingly Cdc48 for recycling. This is not the first paper that argues for a direct endosome to plasma membrane route for recycling proteins (see PMID 15215319) but it does present the most rigorous data I've seen in support of this hypothesis. Similarly, Ist1 was previously implicated in endosomal recycling in animal cells (as opposed to a role in MVB biogenesis like most ESCRT-associated proteins) but observing this requirement in yeast allows application of facile molecular approaches to better understand mechanism. The data implicating Cdc48, its Npl4 adaptor and ubiquitination of Ist1 in its recycling function provides layers of new mechanistic insight. The data presented support a model whereby Npl4 recognition of Ist1-Ub recruits Cdc48 for disassembly of Ist1 oligomers. Precisely how the cargo is selected for transport in this route remains to be determined. I thought the data presented in figures 1 - 3 provide a very rigorous support of a recycling route that does not involve the Sec7 compartment. However, the data in Figure 4 underpinning the new mechanistic insight for how Ist1 works was less convincing.

Primary concerns

1A - The evidence that Ist1 ubiquitination is important in the recycling pathway comes from complementation tests with Ist1^{KR}-HA, which carries more than 20 K to R mutations. Whether it is the loss of ubiquitin modification or a perturbation in some other aspect of Ist1 function is unclear.

A large series of control experiments have been performed to confirm that lack of Ist1 ubiquitination is responsible for defects in recycling, shown by the Ist1^{KR} mutant (Figure 7) and now a DUB-fusion strategy (Figure 9). These valid points were raised by both other reviewers (Reviewer 1, point 2 and Reviewer 2, point 4).

1B - The authors should test Ist1-GFP-DUB and a catalytically dead version of this construct for complementation of ist1 Δ defects in FM4-64 and/or Fur4 recycling.

Great suggestion, this is now included as Figure 9F - 9G and the results fully validate the original model. This alternative approach to removing ubiquitination from Ist1 results in a block in recycling, just like Ist1^{KR} versions and the reliance on ubiquitination is demonstrated using a catalytically dead control version of the DUB-fusion.

2A - The data shown in Figure 4H is minimal support to claim that Ist1 is ubiquitinated, which is a major conclusion reflected in the title of the manuscript. The authors could apply a mass spec approach by using a protease other than trypsin to cleave Ist1 to identify ubiquitinated lysines.

This suggestion led to some very encouraging discussions about alternative proteases that might work. A large series of *in silico* digests using many different enzymes led to our strategy to dilute urea and use ArgC enzyme in a standard 50 mM ammonium bicarbonate buffer, followed by acidification with formic acid to allow a sequential digest using ProAlanase.

Our desire to identify ubiquitinated species of Ist1 also led to the idea of introducing a step to propionylate peptides. All samples were run +/- propionylation and in combination we had impressive coverage of ~5500 peptides from the proteins isolated following two affinity purification steps (Supplemental Table T5).

Despite this, the advance for Ist1 is still only 1/38 lysines confirmed as ubiquitinated (Figure 6F). This definitively shows Ist1 to be ubiquitinated, which is now also better supported by the new ubiquitome purification controls +/- IST1 (Figure 6H). Additionally, the cell biology and proteomic processing strategies we developed to achieve this might be of useful for other proteins that are more tractable for proteomic assessment. We hope this

effort and discussion, in addition to the complementary approaches and various additional controls support the conclusion and title that ubiquitination regulates the role of Ist1 in endosomal recycling.

2B - Conversely, the authors could treat samples comparable to those shown in Fig4H, but isolated under non-denaturing conditions, with or without deubiquitinase and determine if any of the bands detected with Ist1 antibody collapse to a mobility closer to the predicted molecular weight of 34.5 kDa.

We did not follow this line of investigation due to the difficulty of preserving conjugated ubiquitin modifications from proteins generally when performing lysis in non-denaturing conditions (Katzmann & Wendland, *Methods in Enzymology* 2005; MacDonald et al., *Traffic* 2017). Instead, we addressed the alternative suggestions mentioned in (2A) and (2C).

2C - No data is presented here to convince the reader that the bands shown are actually Ist1 - it would be important to pulldown ubiquitinated proteins from an *ist1Δ* strain and probe that sample with anti-Ist1 as a control.

This suggestion also worked - shown in **Figure 6G - 6H** - thank you so much for all these ideas to strengthen the paper.

3. The authors provide good genetic support for the role of Npl4 and Cdc48 in recycling, but the model could be reinforced by determining if they can detect the Cdc48-Npl4-Ist1-Ub interactions (as shown in Fig4I) using pulldown approaches.

We tried this but were unable to immuno-isolate stable proteins from different lysis protocols, never mind perform IPs. In the revised text we are particularly careful describing the model as speculative and explaining what data supports this whilst including alternative views.

4. The data shown in Figures 4L and M are not very convincing to argue for a role of Ist1 ubiquitination and Cdc48-Npl4 in regulating endosome dynamics of Ist1. The data may be statistically significant but this does not equate to biological significance. In addition, the method of quantitation was not described for these imaging data. I suggest either removing these results or try to provide another method of evaluation, such as subcellular fractionation by differential centrifugation followed by western blot for Ist1.

We agree and believe the issues described (reviewer 1, point 3A) regarding mis-localization phenotype make these difficult to interpret clearly. The suggested experiment from this point regarding combination mutants of *ist1* and either *npl4* and *cdc48* did show no additional defects (**Figure 5H - 5I**), which support the original model. We include this previous data regarding Ist1 micrographs as supplemental (much like the mis-localization data) and now do not to over-interpret. We also explain the analysis that was performed in the methods.

5. Figure 4N lacks a catalytically dead control or at least a non-DUB fusion control for comparison.

This is now included as **Figure 9B - 9E** and shows the expected phenotype: colocalizing with Vps4, and not Sec7, much like Ist1^{KR}.

6. Figure 4L shows punctate localization of Ist1KR-GFP while supplemental figure 4d shows that Ist1-GFP-DUB is nearly all cytosolic when expressed in *ist1Δ* cells but punctate when expressed in WT cells. I don't believe the latter result was described or discussed anywhere in the text. Does this imply a role of Ist1 ubiquitination in assembly or recruitment to the endosome? Does a catalytically dead version of this construct localize to punctae in *ist1Δ* cells?

We believe that the Ist1-DUB as sole copy of *IST1* might deubiquitinate cargo at MVBs - much like deletion of the ESCRT-0 ubiquitin receptor *vps27Δ* (Rue et al **MBoC**). As removing ubiquitin-cargo disperses later ESCRTs (MacDonald **EMBO rep** 2012) we assume this explains this localisation result. We felt this observation only served to confuse the reader and did not add useful detail, so have removed it. However, if it was considered beneficial this could be added as a supplement with reference to the Vps27 data from the Emr lab.

Minor comments

a. Figure 1F vs 1G. The text states that the DUB fusion is causing an increased localization of Snc1 and Snc2 to

the Golgi but the MOC vacuoles look about the same. A MOC of 75% seems inflated for GFP-Snc2 relative to the image shown.

This is a good point that I had missed. Qualitatively we observe less surface signal at the PM following DUB-fusion, which has been quantified previously (Xu *et al.*, eLife 2017). We have corrected the text to reflect the analysis and proper conclusions. We do note that most experiments in Figure 1 exhibit a range of MOC values, which is why we felt it necessary to quantify and display all the data as a jitter plot for each. We also now mainly emphasise the key points from these data better, which is the comparison between Sec7 and Vps4 colocalization with this range of recycling cargoes.

b. I believe coloc 2 in Image J calculates a Manders' colocalization coefficient and not an overlap coefficient. Was a different plugin used? If an MCC, how were negative values obtained in Fig 3C? The Methods section indicates a Pearson coefficient was reported except for timelapse. More clarity in the approach taken is needed.

Apologies, we used multiple colocalization methods - one reason was because we had no access to Zen software (which calculates MOC) during lockdowns. We now include thorough description of all methods used.

c. Figure 1 I and J - what was the time of Met or Ura addition?

5 and 15 minutes, respectively. This is now included in the figure itself.

d. The photoconversion experiments are very cool. Do you see the Mup1-Eos red signal in endosomes come back and merge into the green segment of the plasma membrane? It would be nice to see the fate of those molecules.

This is too technically challenging as we do not have good temporal resolution due to photobleaching by the time the pulse-chase and photoconversion has been captured. The few frames we have available after this are insufficient for accurate particle tracking. However, we now include Movie S7 that captured the trafficking of one such event (albeit not photoconverted).

e. The first Intro paragraph covers a lot of material could be split into 2 or 3 paragraphs.

Thank you for this suggestion, it was too dense. I think the flexibility of resubmitting as an article has righted many of these wrongs. We expanded the initial introduction to fully describe eukaryotic surface proteomes (paragraph 1), then split the remainder into degradation (paragraph 2) and recycling (paragraph 3) pathways, all with less focus on only yeast mechanisms. The revised version is much larger, to describe new data but also previous data in more detail, and to include more discussion to the results section.

f. Owing should be owing on the top of page 4.

Got it, thank you!

August 17, 2022

RE: JCB Manuscript #202109137R

Dr. Chris MacDonald
University of York
Department of Biology
Office Biology / J001
Wentworth way
York, --- YO10 5DD
United Kingdom

Dear Dr. MacDonald:

Thank you for submitting your revised manuscript entitled "Recycling of cell surface membrane proteins from yeast endosomes is regulated by ubiquitinated Ist1". The reviewers all now support publication so we would be happy to publish your paper in JCB pending final revisions necessary to meet our formatting guidelines (see details below). We would also like to commend you on the quality of your revisions. In your final text, please be sure to address the reviewers' final minor comments.

A. MANUSCRIPT ORGANIZATION AND FORMATTING:

Submission of a paper that does not conform to JCB guidelines will delay the acceptance of your manuscript.

- 1) Text limits: Character count for Articles is < 40,000, not including spaces. Count includes abstract, introduction, results, discussion, and acknowledgments. Count does not include title page, figure legends, materials and methods, references, tables, or supplemental legends. * Please reformat your final text as an Article with separate results and discussion sections *
- 2) Figures limits: Articles may have up to 10 main text figures.
- 3) Figure formatting: Scale bars must be present on all microscopy images, including inset magnifications. Molecular weight or nucleic acid size markers must be included on all gel electrophoresis.
- 4) Statistical analysis: Error bars on graphic representations of numerical data must be clearly described in the figure legend. The number of independent data points (n) represented in a graph must be indicated in the legend. Statistical methods should be explained in full in the materials and methods. For figures presenting pooled data the statistical measure should be defined in the figure legends. Please also be sure to indicate the statistical tests used in each of your experiments (either in the figure legend itself or in a separate methods section) as well as the parameters of the test (for example, if you ran a t-test, please indicate if it was one- or two-sided, etc.). Also, if you used parametric tests, please indicate if the data distribution was tested for normality (and if so, how). If not, you must state something to the effect that "Data distribution was assumed to be normal but this was not formally tested."
- 5) Abstract and title: The abstract should be no longer than 160 words and should communicate the significance of the paper for a general audience. The title should be less than 100 characters including spaces. Make the title concise but accessible to a general readership.

We typically ask that species name is not included in the title, however here it does seem essential to include a reference to "yeast endosomes"
- 6) * Materials and methods: Should be comprehensive and not simply reference a previous publication for details on how an experiment was performed. Please provide full descriptions in the text for readers who may not have access to referenced manuscripts. *
- 7) Please be sure to provide the sequences for all of your primers/oligos and RNAi constructs in the materials and methods. You must also indicate in the methods the source, species, and catalog numbers (where appropriate) for all of your antibodies. Please also indicate the acquisition and quantification methods for immunoblotting/western blots.
- 8) Microscope image acquisition: The following information must be provided about the acquisition and processing of images:
 - a. Make and model of microscope

- b. Type, magnification, and numerical aperture of the objective lenses
- c. Temperature
- d. Imaging medium
- e. Fluorochromes
- f. Camera make and model
- g. Acquisition software
- h. Any software used for image processing subsequent to data acquisition. Please include details and types of operations involved (e.g., type of deconvolution, 3D reconstitutions, surface or volume rendering, gamma adjustments, etc.).

9) References: There is no limit to the number of references cited in a manuscript. References should be cited parenthetically in the text by author and year of publication. Abbreviate the names of journals according to PubMed. * Supplemental references are not permitted *

10) Supplemental materials: There are strict limits on the allowable amount of supplemental data. Articles may have up to 5 supplemental figures. Please also note that tables, like figures, should be provided as individual, editable files. A summary of all supplemental material should appear at the end of the Materials and methods section.

13) ORCID IDs: ORCID IDs are unique identifiers allowing researchers to create a record of their various scholarly contributions in a single place. At resubmission of your final files, please consider providing an ORCID ID for as many contributing authors as possible.

Please note that JCB now requires authors to submit Source Data used to generate figures containing gels and Western blots with all revised manuscripts. This Source Data consists of fully uncropped and unprocessed images for each gel/blot displayed in the main and supplemental figures. Since your paper includes cropped gel and/or blot images, please be sure to provide one Source Data file for each figure that contains gels and/or blots along with your revised manuscript files. File names for Source Data figures should be alphanumeric without any spaces or special characters (i.e., SourceDataF#, where F# refers to the associated main figure number or SourceDataFS# for those associated with Supplementary figures). The lanes of the gels/blots should be labeled as they are in the associated figure, the place where cropping was applied should be marked (with a box), and molecular weight/size standards should be labeled wherever possible.

B. FINAL FILES:

**It is JCB policy that if requested, original data images must be made available to the editors. Failure to provide original images

upon request will result in unavoidable delays in publication. Please ensure that you have access to all original data images prior to final submission.**

Thank you for this interesting contribution, we look forward to publishing your paper in Journal of Cell Biology.

Sincerely,

Elizabeth Miller, PhD
Monitoring Editor

Andrea L. Marat, PhD
Senior Scientific Editor

Journal of Cell Biology

Reviewer #1 (Comments to the Authors (Required)):

The manuscript by Laidlaw et al is much improved from its initial submission and I would like to commend the authors on their work to address Reviewer's concerns and their thoughtful, courteous, and thorough response. Based on the added figures and extensive work performed, I am satisfied that the authors have made a good faith and rigorous effort to present as complete a story as possible. While the work raises some exciting questions for the future of this pathway and how it contributes to protein sorting, this initial story will be highly valuable to the cell biology community in its current form.

Reviewer #2 (Comments to the Authors (Required)):

The authors have adequately addressed my major concerns in their rebuttal letter and in the improved version of their manuscript.

Minor points:

Figure call out (line 288) is not correct - it should be fig. S5 : Combining the lysineless version of Ist1KR-GFP with Npl4 / Cdc48 mutants results in its mislocalization to the nucleus (Figs. S4C - S4D), which precluded assessment of endosomal recruitment but further alludes to a biological connection between ubiquitinated Ist1 and Npl4-Cdc48.

Perhaps the authors could comment why Ist1-KR protein levels are lower compared to WT? How does that align with the idea that Ist1 is degraded by the proteasome?

Reviewer #3 (Comments to the Authors (Required)):

The authors have done a tremendous amount of work to address the reviewer's concerns and I think the manuscript is substantially improved by their efforts. The data supporting the conclusions are now very strong and the discoveries are

substantial. All of my concerns have been alleviated.

I have one very minor suggestion for improvement. The description of the data shown in Figure 8 and presented on page 5 was a little confusing. I had to read this through a few times to understand what was done and why. My suggestion is to make sure the rationale for the experimental strategy is clearly stated at the beginning of the paragraph. I would also further emphasize that it is the wild-type protein being expressed at lower levels to mimic the expression level of the mutant protein.

We thank the reviewers and editors for final suggestions to make the manuscript better. We now include:

1. The eTOC blurb has been re-written.
2. The references from the supplemental document have been removed and incorporated into the main reference list (to avoid the referencing software causing havoc, these additions have not been tracked in the Word document uploaded).
3. Author contributions have been added using the CRediT system.
4. Small changes have been made to the methods, mainly to avoid the protocol description that had previously only been referenced but additional detail also added to meet specifications (e.g. imaging media).
5. We have moved all the extraneous discussion pieces from the results sections and incorporated them into a more conventional DISCUSSION section. We apologise for the messy nature of the formatting of the RESULTS section, where essentially two styles had been used.
6. Statistical tests are now described in more detail in supplemental table, and the figure legends have been updated to include reference to all replicates, etc.
7. Based on the brief reviewer comments for the resubmitted version we have corrected highlighted typos and modified the text explaining the YETI-Ist1 work for clarification.